# Cryo-EM structures of NHEJ assemblies with nucleosomes

Chloe Hall[1], Philippe Frit [2], Antonia Kefala-Stavridi[3], Amandine Pelletier[4], Steven W. Hardwick [3], Himani Amin [1], Matthew K. Bilyard[5,9], Taiana Maia De Oliviera [6], Ammarah Tariq [1], Sayma Zahid [1], Dimitri Y. Chirgadze [3], Shankar Balasubramanian [5], Katheryn Meek [7], Virginie Ropars[8], Jean-Baptiste Charbonnier [8], Mauro Modesti [4], Patrick Calsou [2], Sébastien Britton [2], Tom L. Blundell [3,10], Thomas Schalch [1] & Amanda K. Chaplin [1] ✉

DNA double-strand breaks (DSBs) are highly deleterious lesions that can trigger cell death or carcinogenesis if unrepaired or misrepaired. In mammals, most DSBs are repaired by non-homologous end joining (NHEJ), which begins when Ku70/80 binds DNA ends and recruits DNA-PKcs to form the DNA-PK holoenzyme. Although recent cryo-EM studies have resolved several NHEJ assemblies, how these factors access DSBs within nucleosomes remains unclear. Here, we present cryo-EM structures of human Ku70/80 and DNA-PK bound to nucleosomes. Ku70/80 binds the DNA end and bends it away from the nucleosome core, while the Ku70 C-terminal SAP domain makes an additional, specific DNA contact. Our DNA-PK–nucleosome structure further reveals the opening of the Ku80 vWA domain, and we show that non-hydrolysable ATP promotes synapsis by stabilising the Ku80-mediated DNA-PK dimer. These structures reveal a model for DSB recognition on nucleosomal DNA and provide insights relevant to targeting NHEJ in cancer therapy.

During the NHEJ DNA-repair mechanism, the DNA break is recognised by the Ku70/80 heterodimer, which subsequently recruits DNA-PKcs to form the DNA-dependent protein kinase (DNA-PK) holoenzyme complex at the DNA damage site. It has been shown through recent cryo-EM structures and single-molecule experiments that following assembly of the DNA-PK holoenzyme, a long-range synaptic complex (LRC) is assembled on the broken DNA ends. There are at least two forms of LRCs, composed of dimers of DNA-PK, either mediated by XLF or the C-terminus of Ku80[1–4]. These DNA-PK LRCs both hold the DNA ends in a pre-synaptic state at ~115 Å apart. The DNA ends may then require further processing by nucleases and polymerases to allow them to be compatible for ligation. It has recently been shown through mutational analysis that the two LRC DNA-PK dimers have unique roles in either promoting fill-in end processing or DNA end resection[5]. Finally, DNA-PKcs is phosphorylated and removed from the complex, allowing for a short-range synaptic complex (SRC) to be formed, where

[1]Leicester Institute for Structural and Chemical Biology, Department of Molecular and Cell Biology, University of Leicester, Leicester, UK. [2]Institut de Pharmacologie et Biologie Structurale (IPBS), Université de Toulouse, CNRS, Université Toulouse III - Paul Sabatier (UT3), Toulouse, France. [3]Department of Biochemistry, University of Cambridge; Sanger Building, Cambridge, UK. [4]Cancer Research Center of Marseille, Department of Genome Integrity, CNRS UMR7258, Inserm U1068, Institut Paoli-Calmettes, Aix Marseille Univ, Marseille, France. [5]Department of Chemistry, University of Cambridge, Cambridge, UK. [6]AstraZeneca R&D, Discovery Sciences, Mechanistic and Structural Biology, Cambridge, UK. [7]College of Veterinary Medicine, Department of Microbiology & Molecular Genetics, Department of Pathobiology & Diagnostic Investigation, Michigan State University, East Lansing, MI, USA. [8]Université Paris-Saclay, CEA, CNRS, Institute for Integrative Biology of the Cell (I2BC), Gif-sur-Yvette, France. [9]Present address: Pharma Research and Early Development, Roche Innovation Center Basel, F. Hoffmann-La Roche Ltd, Basel, Switzerland. [10]Present address: Department of Medicine, University of Cambridge, Heart and Lung Research Institute (HLRI), Cambridge, UK. ✉e-mail: ac853@leicester.ac.uk

the Ligase IV catalytic domain can engage and ligate the DNA ends, thus completing the DNA repair process[3].

All structures of the NHEJ machinery to date have been solved with "free" short pieces of linear double-stranded DNA. However, in eukaryotes, DNA in the nucleus is compacted and packaged within chromatin, where nucleosomes are the fundamental structural unit. Nucleosomes comprise two copies of each histone H2A, H2B, H3 and H4, together assembling a histone octamer with a basic outer surface, which organises 147 bp of DNA into 1.65 turns of a left-handed superhelix, forming the nucleosome core particle (NCP)[6]. Nucleosomes array on eukaryotic genomes like beads on a string, and DSBs will result in chromatinized DNA ends. Several studies have identified PARP-dependent chromatin remodelling as stimulating the early steps of NHEJ, including Ku70/80 loading, suggesting that chromatin remodellers such as ALC1/CHD1L, CHD2 or CHD7 could be required for Ku70/80 and the rest of the NHEJ machinery to associate with DSBs within chromatinized DNA[7–10]. However, these mechanisms and the ability of Ku70/80 and DNA-PKcs to promote repair in the context of chromatin remain unclear.

## Results and discussion

### The architecture of Ku70/80 binding to nucleosomes

To determine whether Ku70/80 can bind to DNA when packaged within a nucleosome, we designed a nucleosome construct containing the minimum DNA length of 147 bp to encircle the histone octamer. We first purified Ku70/80 from insect cells to a high purity (Supplementary Fig. 1) and reconstituted 147 bp nucleosomes. Using electromobility shift assays (EMSAs) we were able to confirm the binding of Ku70/80 to the nucleosome, with a stable band present at a ratio of 1:4 (nucleosome: Ku70/80) (Supplementary Fig. 2A). We then prepared cryo-EM grids of this complex sample and following data collection and processing (Supplementary Fig. 3 and Table 1), we produced a map to a global resolution of 3.3 Å (Fig. 1B). We could confidently dock a nucleosome with the 147 bp Widom "601" positioning sequence and a single Ku70/80 molecule. The modelled Ku70/80 molecule is positioned such that the DNA from the nucleosome enters through the preformed ring of Ku70 and Ku80 encircling the duplex DNA. The structure of Ku70/80 when bound to the nucleosome is indistinguishable from previous structural studies of Ku70/80 in complex

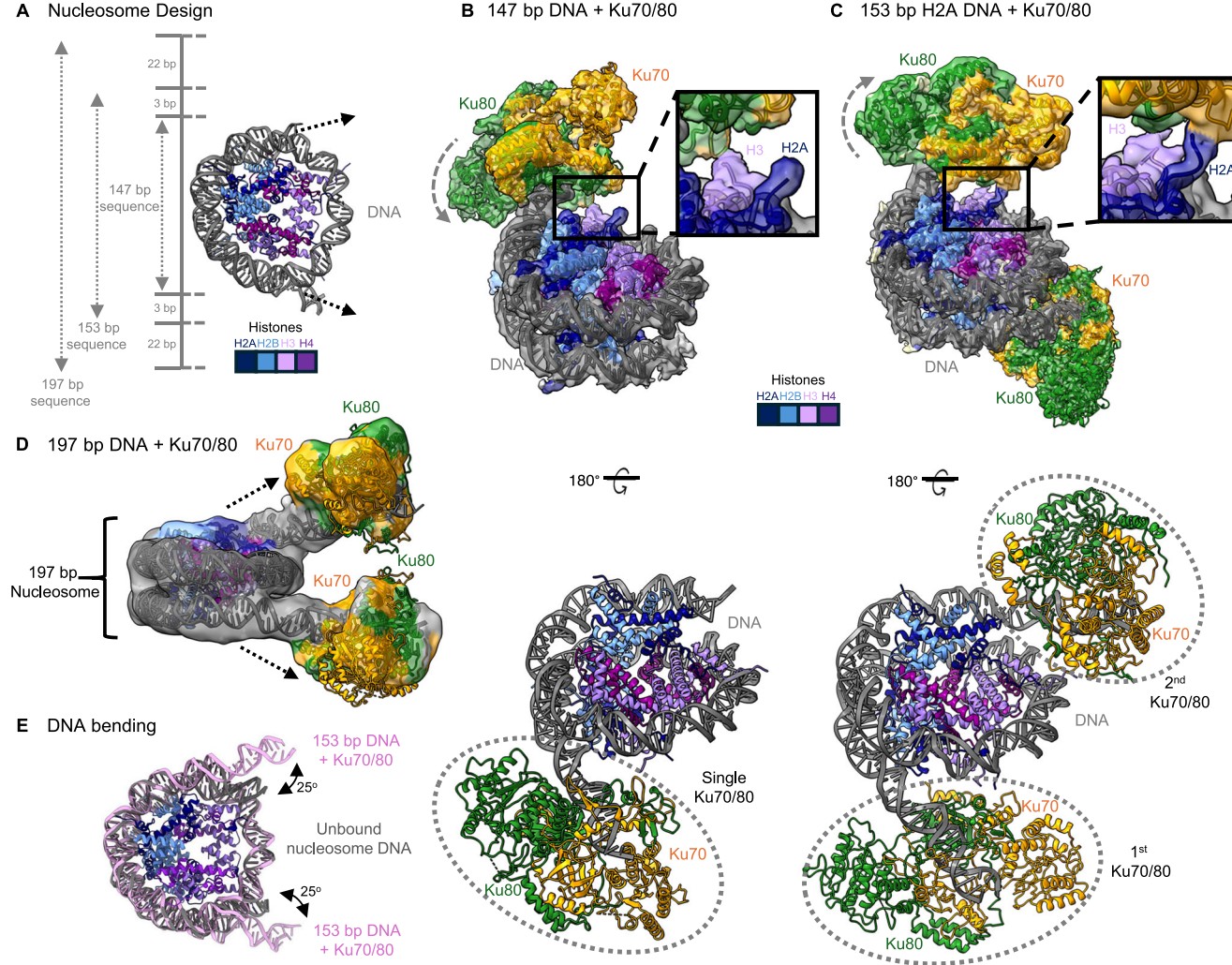

**Fig. 1 | Cryo-EM structures of Ku70/80 bound to nucleosomes. A** Scheme showing the nucleosome design, with the corresponding increase in DNA bp length for 147, 153 and 197 bp. **B** Cryo-EM map and model for Ku70/80 bound to a 147 bp nucleosome to 3.3 Å resolution, with a 180° view of the model highlighting Ku70/80 DNA binding. Inset, showing H3 and H2A orientations with respect to Ku70/80. **C** Cryo-EM map and model for two Ku70/80 molecules binding to a 153 bp nucleosome to 3.5 Å resolution, with a 180° view of the model highlighting orientations of the two Ku70/80 molecules binding to DNA. Inset, showing H3 and H2A

orientations with respect to Ku70/80. **D** Cryo-EM map and model for two Ku70/80 molecules bound to 197 bp nucleosome, arrows showing Ku70/80 bound to the DNA ends. **E** Overlaid model of PDB 6JXD (containing the usual Widom "601" positioning sequence) with no protein bound to the DNA in grey, and DNA from the 153 bp from (**C**) shown in pink, highlighting the bending of the DNA by 25°. In the structures, Ku70 is in orange, Ku80 in green, DNA in dim grey, H2A in navy, H2B cornflower blue, H3 medium purple and H4 purple.

with short lengths of DNA as shown in (Supplementary Fig. 7A). Ku70/80 covers approximately 10 bp of DNA when bound in this structure, as calculated using PISA analysis[11] (Supplementary Fig. 8A). This model therefore confirms that Ku70/80 can bind nucleosomal DNA in the absence of DNA flanking the nucleosome.

Within our initial nucleosome: Ku70/80 map, there was weak density visible corresponding to a second Ku70/80 molecule bound on the opposite side of the nucleosome on the other exposed DNA end. We may be observing a mixture of Ku70/80 bound to either DNA end in our dataset, which is being averaged out. In an attempt to resolve the density for this second Ku70/80 molecule, we increased the length of the DNA to 153 bp. Following grid preparation of Ku70/80 with the 153 bp nucleosome, data collection and processing, we resolved a cryo-EM map to an overall resolution of 3.5 Å. Within this map, we could model two Ku70/80 molecules within the density present on both DNA ends (Fig. 1C). Interestingly, in the 147 bp nucleosome map, the N-terminus of H3 could be seen coming into close proximity with Ku70/80. Similarly, H3 appears to directly contact Ku70/80 when the DNA length is increased to 153 bp. This H3 interaction site on Ku80 is the same region that interacts with the Ligase IV BRCT domain in the SRC (Supplementary Fig. 9). In addition to H3, the C-terminal tail of H2A appears to contact Ku70/80 only when the DNA length is increased to 153 bp (Fig. 1C). Although interactions between the histone tails and Ku70/80 were observed, direct binding between peptides, both with and without different post-translational modifications, and Ku70/80 was not detected in assays conducted by EpiCypher. In this assay, 280 biotinylated histone peptides were tested to see if they interact with Ku70/80. Interactions between the histone tails observed and Ku70/80 were not detected, and only potential interactions with histone tails far from the site of interaction were observed (Supplementary Fig. 10). These interactions may therefore be weak and only present in the context of nucleosome-Ku70/80 binding. The small increase in DNA length (3 bp to each DNA end) causes Ku70/80 to rotate, indicating its ability to thread onto the DNA in a corkscrew-like manner to some extent. However, like that identified in the 147 bp Ku70/80 structure, the Ku70/80 molecules were also identical in structural orientation to those previously reported and covered ~10 bp of DNA[12] (Supplementary Figs. 7B and 8B).

To determine whether Ku70/80 can continue to translocate inwards towards the nucleosome core upon increasing the DNA length (distance of the break from the NCP), we increased the DNA length to 197 bp. In this cryo-EM map (solved to an overall lower resolution of 7.6 Å), we could visualise the increase in the DNA length and two Ku70/80 molecules binding at the ends of the DNA chain (Fig. 1D). The DNA from the nucleosome still enters through the preformed ring of Ku70 and Ku80 encircling the duplex DNA, covering ~12 bp of DNA (Supplementary Fig. 8C). The limited resolution in this map is likely due to the flexibility of the Ku70/80 molecules on the extended DNA. However, the Ku70/80 molecules are clearly positioned distant to the NCP at the DNA termini, indicating that the corkscrew-like threading of Ku70/80 seen when increasing the DNA length from 147 to 153 is limited, and that Ku70/80 is unable to translocate along the DNA chain progressively. We can also again recognise that the Ku70/80 molecules are similar in structure to those previously determined[12] (Supplementary Fig. 7C). From the three varied DNA length nucleosome constructs, it appears that Ku70/80 does not have a preference for linker length.

Comparing the models generated from the three different nucleosome DNA lengths, it must first be noted that the mode of Ku70/80 binding is identical in all three nucleosome constructs, with the DNA binding to the heterodimer as previously determined[12] (Supplementary Fig. 7A–C). It can also be seen that in the 147 and 153 bp DNA, binding of Ku70/80 causes the DNA to bend away from the NCP by approximately 25° to allow threading of Ku70/80 (Fig. 1E and Supplementary Fig. 11). This corroborates previous studies suggesting

"peeling" of the DNA away from the NCP upon Ku70/80 binding[13]. This may also be caused by unwrapping of the nucleosomal DNA that allows Ku70/80 to capture the nucleosomal DNA ends[14,15]. However, when the DNA length was increased to 197 bp, the DNA followed its canonical path on the NCP. This indicates that the peeling of the DNA is caused by a steric hindrance, and once Ku70/80 captures the DNA ends, they need to be displaced from the histone octamer. However, once the DNA length is increased, the DNA ends are more readily accessible/flexible to Ku70/80 binding and do not require peeling away from the NCP.

In all our cryo-EM maps of Ku70/80 with nucleosomes, we do not observe multiple copies of Ku70/80 loading onto a single DNA end. It has been shown that multiple Ku70/80 molecules can load onto linear DNA in vitro; however, this is limited to -1–2 molecules in cells[14–17]. It has recently been shown that DNA-PKcs is responsible for preventing multiple loading of Ku70/80 onto chromatin. Ku70/80 accumulation is further restricted by a neddylation/FBXL12-dependent process that actively removes loaded Ku70/80 molecules throughout the cell cycle and a CtIP/ATM-dependent mechanism that operates in S-phase[18]. Likely, we do not see multiple Ku70/80 molecules loading in our cryo-EM data, as this is sterically inhibited by the relatively limited length of DNA used. Additionally, it could be considered that the ratio used in our cryo-EM experiments (1:4, nucleosome: Ku70/80) is not permissive, or the loading of multiple Ku70/80 molecules requires active chromatin remodelling.

## The Ku70 SAP domain directly binds to the nucleosomal DNA

Phosphorylation of the H2AX variant is a well-known early step in the response to DNA damage, which can be performed by the kinase activity of DNA-PKcs. It is known to primarily phosphorylate H2AX at serine 139. To assess whether the presence of H2AX in the nucleosome alters the binding activity of Ku70/80, we reconstituted the 153 bp nucleosome with H2AX replacing canonical H2A (Supplementary Fig. 12) and, similar to the H2A-containing nucleosomes, we performed binding and structural assays. As with canonical H2A, EMSA gels with H2AX containing nucleosomes confirmed the formation of a stable Ku70/80-nucleosome complex at a 1:4 (nucleosome: Ku70/80) ratio (Fig. 2A). Following cryo-EM data collection and data processing, two cryo-EM maps were produced, one corresponding to the 153 bp nucleosome bound to two molecules of Ku70/80 in a similar arrangement to the 153 bp nucleosome containing canonical H2A (Fig. 2B). However, the second map contained additional density corresponding to the Ku70 C-terminal linker and SAP domain (Fig. 2C) (to overall resolutions of 2.9 and 3.4 Å, respectively). The Ku70 C-terminal linker leads down from the core of Ku70/80 and appears to engage with the negatively charged (acidic) patch formed by histones H2AX-H2B (Fig. 2B). This acidic patch is a characteristic feature of the nucleosome surface and is a common site of attachment of various regulatory chromatin proteins[19]. When compared to known H2A binders such as Dot1L, SAGA DUB and COMPASS, a clear overlap can be observed between the linker of Ku70 and these protein complexes. However, it is not clear whether any specific residues overlap between the structures (Supplementary Fig. 13). The SAP domain itself binds on top of the nucleosomal DNA, with residues K596 and T577. It should be noted that we did previously observe some density corresponding to the Ku70 linker and SAP domain in the 153 bp H2A nucleosome datasets, highlighting that this density is not specific to H2AX (Supplementary Fig. 14). The density within the H2A nucleosome map is however, not as easily resolved as in the H2AX dataset, which may be due to data quality or stabilisation of the flexible tail of H2AX. We modelled the linker and SAP domain of Ku70 by docking a known structure of the SAP domain and tracing the linker density up to the core of Ku70. The density for this region of Ku70 is low resolution and was difficult to confidently model amino acid side chains. Notably, AlphaFold 3[20] was unable to accurately predict the position of the

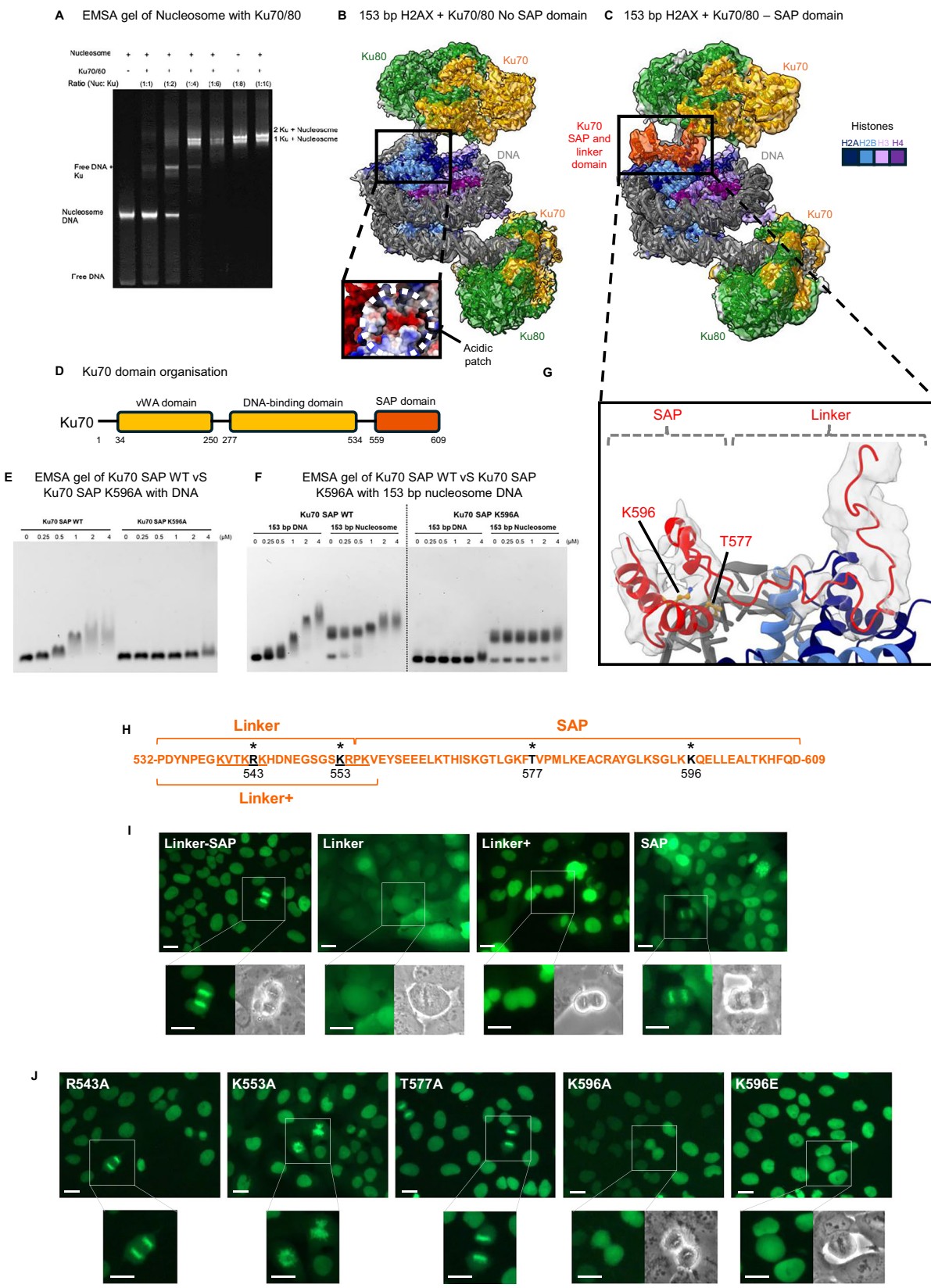

Ku70 SAP domain or linker region (Supplementary Figs. 15 and 16). Similar to the nucleosome construct with 153 bp and canonical H2A (Fig. 1C), the H3 N-terminal and H2AX C-terminal density can be seen to lead towards Ku70/80. However, no additional density was observed that would indicate a specific interaction with Ku70/80 through these histone tails.

To determine the effect of the SAP domain of Ku70 binding to DNA, we also carried out EMSA studies comparing full-length Ku70/80 binding to DNA and Ku70/80-ΔSAP (see "Methods") binding to DNA (Supplementary Fig. 17A). This EMSA gel shows that in the absence of the SAP domain, a higher band shift can be observed. This suggests that the SAP domain controls multiple loading of Ku70/80 onto DNA,

**Fig. 2 | Interaction between the Ku70 C-terminal linker and SAP domain with chromatin.** **A** EMSA gel of a 153 bp nucleosome containing H2AX, with increasing concentrations of Ku70/80, showing a gel shift indicative of binding, DNA bands labelled. This was repeated three times giving the same results. **B** Cryo-EM structure and model of 153 bp H2AX + Ku70/80 with no Ku70 SAP domain or linker present. The inset shows the acidic patch on the histone octamer. **C** Cryo-EM structure and model of 153 bp H2AX + Ku70/80 with the Ku70 SAP domain and linker present. Ku70 is in orange, Ku80 in green, DNA in a dim grey, the SAP domain and linker in dark orange, and histones are coloured according to the key. **D** Ku70 domain organisation, Ku70 in orange and the SAP domain in dark orange. **E** EMSA gel detecting retardation of a 400 bp DNA fragment by purified WT or K596A Ku70 SAP domain in isolation (residues 539–609), indicative of a direct interaction with DNA stabilised by residue

K596. This was one of three replicates. **F** EMSA gel detecting retardation of a 153 bp DNA fragment and a 153 bp nucleosome by purified WT or K596A Ku70 SAP domain in isolation. This was one of three replicates. **G** Close-up of the Ku70 SAP domain and linker, with key residues (K596 and T577) labelled. **H** Amino acid sequence of Ku70 C-terminus showing Linker, Linker+ and SAP domain boundaries. Residues interacting with the nucleosome (see **F** inset) are highlighted with asterisks, and their positions are indicated. Underlined residues correspond to the Nuclear Localization Sequence. **I** Fluorescence imaging of U2OS cells expressing GFP-fused Ku70 domains, as indicated. Insets show enlarged views of mitotic cells by fluorescence and phase contrast imaging. 2–4 Images were taken. **J** Fluorescence imaging, as in (**H**), of U2OS cells expressing mutated Ku70 linker-SAP domains. Scale bars on (**I** and **J**) are 20 μm. Three independent series of photos for the five mutants.

and in its absence, multiple copies of Ku70/80 are able to bind. This is similar to what has recently been observed with the removal of the SAP domain[21]. Furthermore, we also utilised AlphaFold 3[20], to analyse the SAP domain binding to a 20 bp DNA duplex. Interestingly, AlphaFold 3 also predicts the binding of the SAP domain to DNA with K596 interacting directly with the DNA (Supplementary Fig. 17B), as we see in our cryo-EM structure (see further details below). It also shows that the overall SAP domain shows a positively charged patch which binds to the negatively charged DNA (Supplementary Fig. 17D). Furthermore, the Ku70 SAP domain in isolation interacts directly but weakly with a 400 bp DNA fragment in a manner that is stabilised specifically by residues K596 and T577 as evidenced by EMSA analysis (apparent $K_D$ in the 0.5 to 1 mM range). WT vs K596 (Fig. 2E) and WT vs T577 (Supplementary Fig. 17F), therefore showing the importance of these residues in this interaction. Additionally, EMSAs comparing WT SAP and K596A SAP binding to 153 bp free DNA and 153 bp nucleosome-bound DNA demonstrate that the SAP domain alone is capable of interacting with both DNA substrates (Fig. 2F). The SAP domain displays a clear preference for free DNA, as evidenced by the more pronounced mobility shift observed in the corresponding EMSA gels. Consistent with the results obtained using 400 bp free DNA, the K596A mutation markedly reduces the SAP domain's binding affinity for both free and nucleosome-bound DNA. Collectively, these results indicate that the SAP domain can associate with both free and nucleosome-bound DNA, and that residue K596 is critical for this interaction.

## The SAP domain of Ku70 contacts chromatin in vivo

To determine the importance of the interaction between the Ku70 linker and/or SAP domain with chromatin in vivo, we expressed in human U2OS cells these domains fused to GFP, in the absence of the rest of the Ku70 protein. The linker-SAP region of Ku70 (residues 532–609) exhibits a nuclear tropism, likely driven by the presence of the Ku70 Nuclear Localisation Signal (NLS) (Fig. 2H)[22], as well as a high affinity for chromatin with strong chromosome staining in mitotic cells. The linker alone (residues 532–554), which lacks part of the NLS, partially leaks into the cytoplasm and loses its strong binding to chromatin. Conversely, the SAP domain (residues 555–609), despite lacking most of the NLS and partially leaking into the cytoplasm, retains its high affinity for chromatin. To assess the role of the very C-terminal part of the NLS present in the SAP construct, a slightly extended version of the linker domain (linker+: residues 532–557) retaining the entire NLS was tested. Although this construct now localises mainly in the nucleus, it still displays no particular binding to chromatin, suggesting that within the linker-SAP region of Ku70, the SAP domain per se is sufficient to support this strong binding. To further characterise the SAP/chromatin interaction, we mutated the linker-SAP region of Ku70 on residues identified by cryo-EM as potentially interacting with nucleosomes (Fig. 2G). As mentioned above, these residues were difficult to confidently model due to low resolution, but were based on the most likely interactions as compared to previous proteins interacting with the acidic patch. However, in agreement with the above results, mutations on residues R543 and K553 (contacting the acidic patch) lying in the linker domain have no effect on

chromatin binding. On the other hand, although the T577A mutation has no effect either, mutations on K596 (K596A and K596E) completely abolish the strong binding of the Linker-SAP domain to chromatin (Fig. 2I, J). These results suggest that the SAP domain of Ku70 contacts chromatin in vivo in a manner similar to the interaction identified by cryo-EM between Ku70/80 and nucleosome DNA. We therefore propose that the SAP domain is important for DNA binding, whether that be free DNA or packaged within nucleosomes, while the linker region mediates more nonspecific interactions that may vary depending on the DSB site.

Recent studies investigated the role of the Ku70-SAP domain. It has been shown that the Ku70-SAP domain is important in restricting Ku70/80 rotation and lateral movement on DNA, which is largely masked by DNA-PKcs[21]. Using a mouse model, it was shown that deletion of the Ku70 SAP domain did not affect V(D)J recombination or animal development but did impair the animals' and cells' ability to repair exogenously induced DSBs[23]. The SAP mutant cells were unable to retain Ligase IV and additional NHEJ factors on chromatin. Therefore, our structural and functional insights highlight the role and importance of the Ku70 SAP domain in Ku70/80 binding to DSBs in the cell.

## Architecture of DNA-PK binding to nucleosomes

It has been shown recently through measurements in living cells that Ku70/80 and DNA-PKcs arrive at the broken DNA ends simultaneously[24]. Therefore, we next wanted to explore the recruitment of DNA-PKcs to form the DNA-PK holoenzyme complex. Following successful purification of DNA-PKcs from HeLa nuclear extracts (Supplementary Fig. 1), we collected cryo-EM data of Ku70/80 and DNA-PKcs together with a 153 bp nucleosome (containing H2AX) in a ratio of 1:4:2 (DNA: Ku70/80: DNA-PKcs, shown to work with linear DNA previously[2]). Following processing of this dataset (Supplementary Fig. 18 and Table 2), we generated three distinct cryo-EM maps (Fig. 3). The first map (solved to a global 4.5 Å resolution) and corresponding model (Fig. 3A) show Ku70/80 bound to the nucleosome, with density corresponding to DNA-PKcs on the opposite side of Ku70/80 to form the DNA-PK holoenzyme monomer. In this structure, the position of the nucleosome relative to Ku70/80 has changed compared to our structures of Ku70/80 bound to nucleosomes in the absence of DNA-PKcs and is closer to Ku70 (Fig. 3A). In this map, the vWA domain of Ku80 is in a closed conformation, whilst the Ku70 vWA domain is slightly open, as seen previously for DNA-PK cryo-EM structures[2]. The second map (solved to 4.4 Å resolution) also shows a DNA-PK monomer, but here the nucleosome is engaged further away from Ku70, in a similar orientation to Ku70/80 bound to a 153 bp nucleosome in the absence of DNA-PKcs. Ku70 is open to a similar degree as seen in model/map 1 and previous DNA-PK structures; however, in this structure, the Ku80 vWA domain is in an "open" conformation, which has not previously been identified without the addition of XLF or other accessory NHEJ protein factors[25] (Fig. 3B and Supplementary Fig. 22). It should be noted that Ku70/80 binds to the nucleosomal DNA in a similar manner in these structures when compared to Ku70/80 alone, with the DNA entering the central DNA binding pocket. Again, approximately 13 bp are within the Ku70/80 DNA binding pocket,

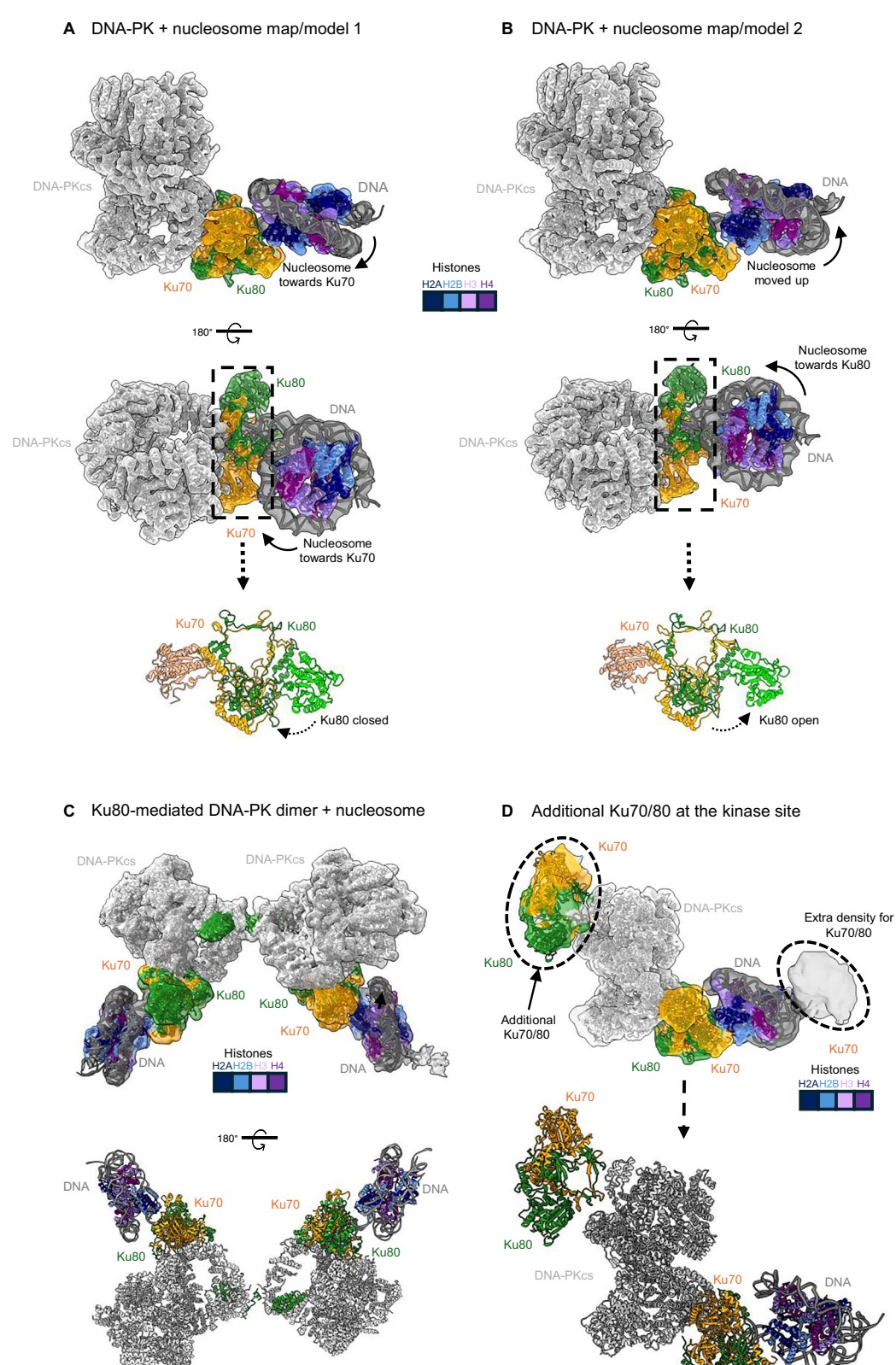

**Fig. 3 | DNA-PK bound to mono-nucleosomes. A** DNA-PK with 153 bp H2AX nucleosome model/map 1 (to 4.5 Å resolution), showing DNA-PKcs and Ku70/80 bound to nucleosome DNA. The nucleosome is positioned close to Ku70 and Ku80 vWA is closed. Ku80 vWA coloured in lime and Ku70 vWA in salmon. **B** DNA-PK with 153 bp H2AX nucleosome model/map 2 (to 4.2 Å resolution), showing DNA-PKcs and Ku70/80 bound to nucleosome DNA. The nucleosome is positioned closer to Ku80 and Ku80 vWA is open. Ku80 vWA coloured in lime and Ku70 vWA in salmon. **C** Ku80-mediated DNA-PK dimer to 4.4 Å resolution, with two nucleosomes held by the formation of the DNA-PK dimer. **D** DNA-PK model from A, but showing a local filtered map with the contour level high, indicating the presence of extra density for an additional Ku70/80 molecule near the DNA-PKcs kinase active site and an additional Ku70/80 at the other side of the nucleosome. Ku70 is shown in orange, Ku80 in green, DNA in dark grey, DNA-PKcs in light grey and histone coloured according to the key.

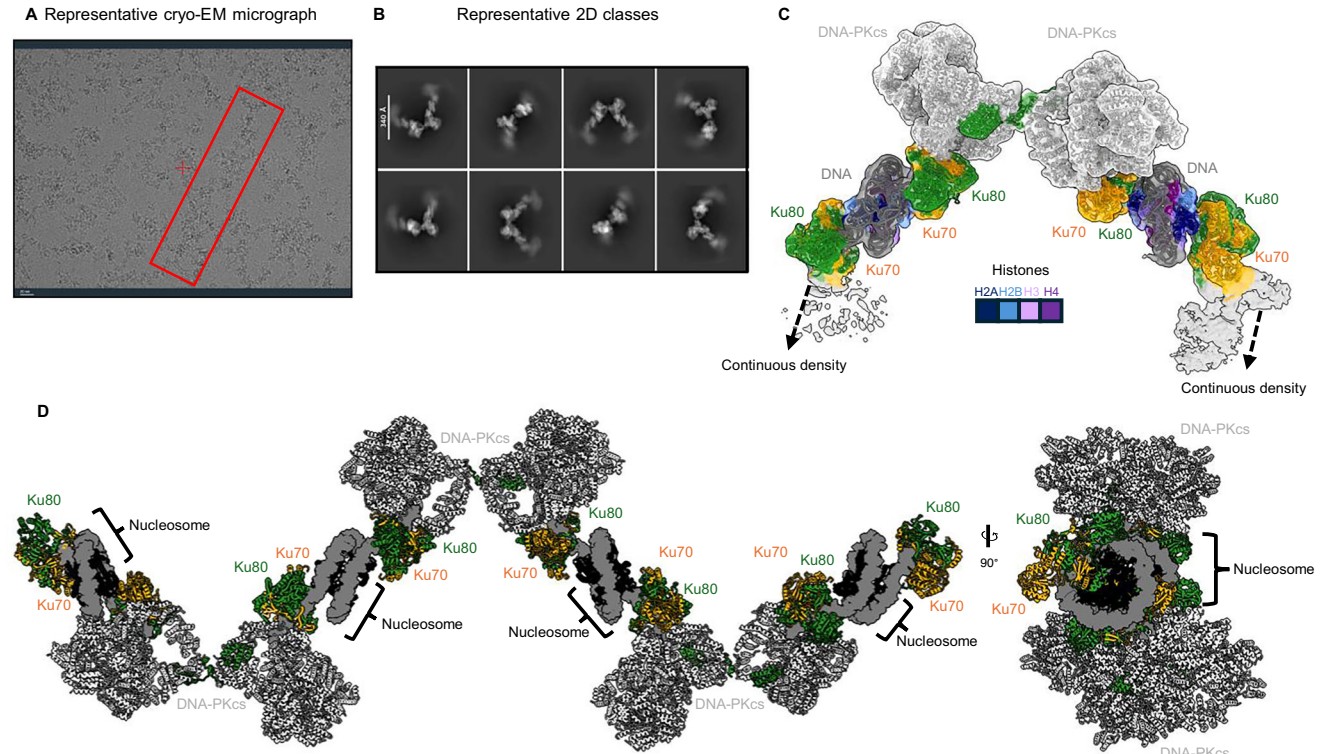

**Fig. 4 | DNA-PK bound to nucleosomes following the addition of AMP-PNP.**
**A** Representative cryo-EM micrograph (from 12,898 micrographs total), scale bar 20 μm, with DNA-PK nucleosome filaments highlighted with a red box.
**B** Representative 2D classes, scale bar 340 Å. **C** Cryo-EM map and model for DNA-PK with 153 bp H2AX nucleosome following the addition of AMP-PNP to 5 Å resolution. DNA-PKcs is shown in light grey, DNA in dark grey, Ku70 in orange, Ku80 in green and the histones coloured according to the key. **D** Model of an extended filament of DNA-PK bound to nucleosomes. Ku70 is shown in orange, Ku80 in green, DNA as a dark grey surface, DNA-PKcs in light grey and the histones in black.

whereas only ~8 bp are covered by DNA-PKcs, likely due to the short length of DNA (Supplementary Fig. 23). The nucleosome conformation also, as mentioned, differs between model 1 and model 2, with this movement coinciding with the opening of the vWA domain of Ku80 (Supplementary Fig. 7D, E). Furthermore, the observed interactions between the histone tails and Ku70/80 are no longer clearly visualised. Remarkably, we no longer observe density for the C-terminal linker and SAP domain of Ku70 within these DNA-PK nucleosome maps. We do, however, regularly observe density at the bottom of the circular cradle of DNA-PKcs, which we predict is the SAP domain of Ku70 (Supplementary Fig. 7G). Therefore, the SAP domain is able to relocate from the NCP to DNA-PKcs upon formation of DNA-PK.

Finally, the third map from this dataset shows a Ku80-mediated DNA-PK dimer (solved to 4.4 Å resolution) (Fig. 3C). This shows the same overall Ku80-mediated dimer structure previously identified[2], but now with a nucleosome on either side of Ku70/80 within this dimeric assembly. In this structure, Ku70 vWA displays the same open conformation as previously shown, with Ku80 adopting an intermediate open state when compared to the monomeric DNA-PK maps[2] (Supplementary Fig. 22). From direct comparisons with model 1 and model 2 DNA-PK monomer nucleosome structures, the DNA-PK dimer model appears more similar to model 2, where Ku70 and Ku80 are open. You can clearly see that in model 1, the NCP appears to have moved compared to model 2 and the DNA-PK dimer (Supplementary Fig. 24). Like with the monomeric DNA-PK models, density is no longer observed for the Ku70 C-terminal linker and SAP domain engaged with the NCP, however, unlike with the DNA-PK monomers we cannot observe any density on DNA-PKcs corresponding to the SAP domain.

Notably, within this DNA-PK nucleosome dataset, we can identify an additional low-population conformation with density extending from the head domain of DNA-PKcs, near the kinase active site

(Fig. 3D). Even though this density is low resolution, we can confidently dock into this a molecule of Ku70/80. The Ku70/80 molecule is positioned such that the bridge region of Ku70/80 is close to the kinase site of DNA-PKcs. This positioning of an additional Ku70/80 molecule has not been previously observed, and we hypothesise that this additional Ku70/80 may be primed for phosphorylation by DNA-PKcs.

## Non-hydrolysable ATP stabilises the Ku80-mediated DNA-PK dimer

After obtaining maps of DNA-PK bound to nucleosomes, we subsequently sought to activate DNA-PKcs by the addition of ATP (grids frozen after ~15 min and 1 h post-ATP), to determine if any structural changes are induced upon kinase activation when bound to nucleosome DNA. Surprisingly, following cryo-EM data collection and processing, we again were able to solve three maps identical to those identified when no ATP was added to the sample (Supplementary Fig. 25). This may be due to the fast turnover of ATP and the absence of further NHEJ proteins such as Ligase IV, XRCC4 and XLF, which are required for the DNA repair mechanism to proceed. To therefore "trap" DNA-PKcs, we added the non-hydrolysable ATP analogue AMP-PNP (see "Methods") to DNA-PK with nucleosomes. Strikingly, we could observe large filaments on the micrographs (Fig. 4A), not seen with samples of only DNA-PK and nucleosome, or when ATP was added. Following cryo-EM data processing, we could observe extended repeating units within the 2D class averages (Fig. 4B), producing a final cryo-EM map to a global resolution of 5 Å (Fig. 4C). In this map, we were able to model the Ku80-mediated DNA-PK dimer bound to a nucleosome on either side (Fig. 3C), with an additional Ku70/80 molecule on either side of the nucleosome. We could observe that ~5 bp and 15 bp enter DNA-PKcs and Ku70/80, respectively, as shown in the previous structures, and ~17 bp in the additional Ku70/80 molecule

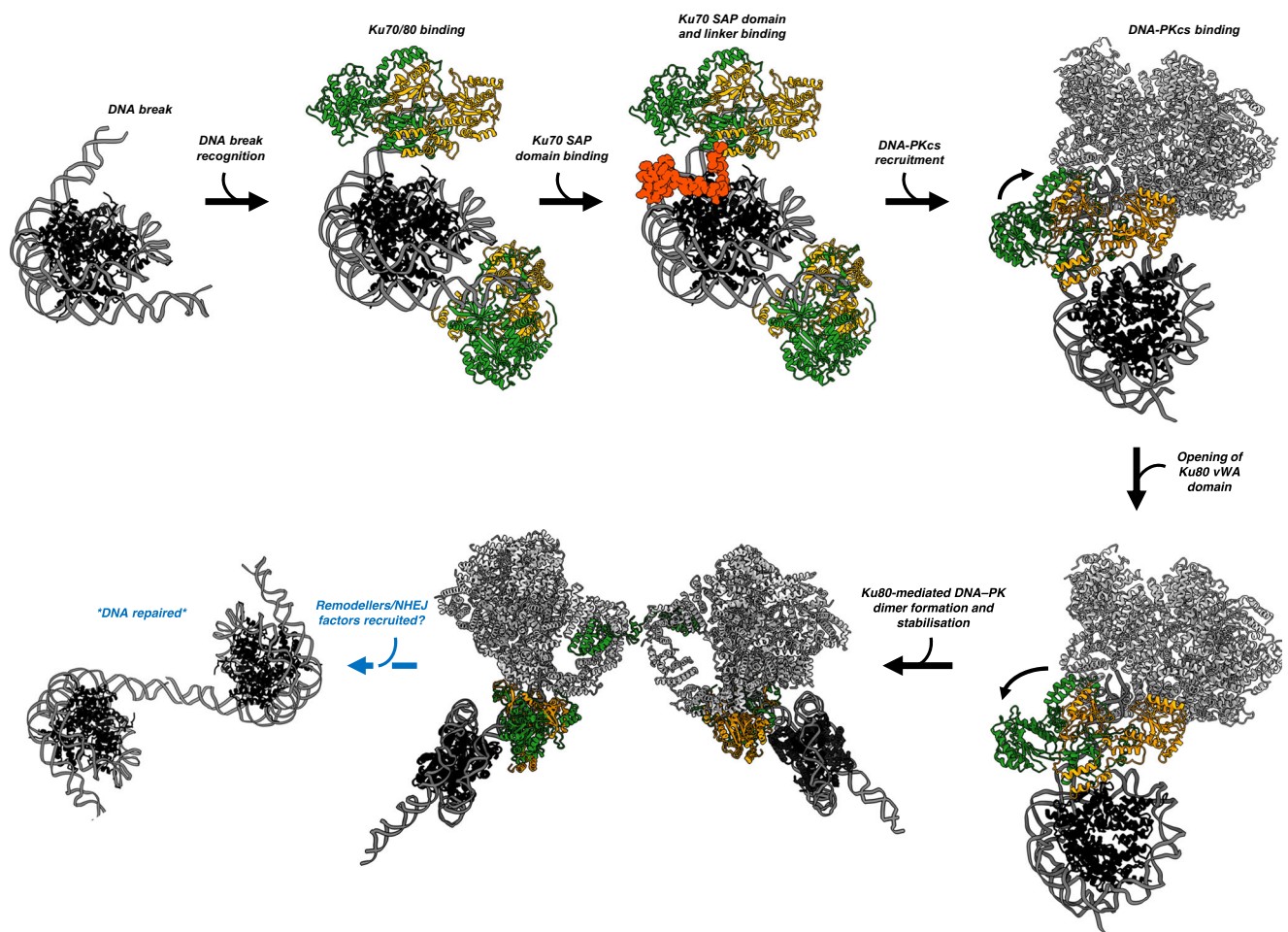

**Fig. 5 | Model of NHEJ DNA repair in the context of nucleosomes.** Histones are coloured in black, DNA in dim grey, DNA-PKcs in light grey, Ku70 in orange, Ku80 in green and the Ku70 C-terminal linker and SAP domain in red.

(Supplementary Fig. 26). We can also observe extended map density for another DNA-PKcs molecule on either side to create an extended filament bridged by bound nucleosomes. We created a model of this extended filament of DNA-PK and nucleosomes based on our cryo-EM data. In this experimental map, we do not observe density for the additional Ku70/80 molecule close to the kinase site, as shown in Fig. 3. We do not observe clear density for AMP-PNP in the active site, which has been shown previously[26]. This may be due to the relatively modest resolution of this map. The repeating unit of this filament is very similar to the DNA-PK dimer model shown in Fig. 3C. It is apparent, however, that the addition of AMP-PNP results in a striking structural stabilisation. We propose that the stabilised filament structure observed upon the addition of AMP-PNP reflects that this trapped state favours the formation of the Ku80-mediated LRC and ultimately should enhance DNA synapsis. This may explain how DNA-PK quickly couples synapsis with kinase activation, leading to more efficient DNA repair.

## Chromatin remodelling and DNA end ligation
Attempts to solve structures of more complex NHEJ machineries in complex with nucleosomes (with the addition of Ligase IV, XRCC4 and XLF) proved unsuccessful, resulting in maps that represented only sub-assemblies of this machinery. For example, using the 153 bp nucleosome construct produced only maps of DNA-PK bound to nucleosomes. However, this is perhaps not surprising, as when we docked the XLF-mediated LRC onto our cryo-EM structures, we observed a steric clash with the stalk of XRCC4 and BRCT domains of Ligase IV with the

nucleosome (Supplementary Fig. 27). We subsequently extended the DNA, but could still only visualise DNA-PK bound to nucleosomes. While it may be that further optimisation of this complex is necessary, we additionally hypothesise that a subsequent remodelling step of the nucleosome DNA is required before further NHEJ proteins can be engaged. It has been recently shown that the Artemis: DNA-PKcs nuclease complex does not act on mono-nucleosome substrates, highlighting the steric challenges posed by nucleosomes[27].

## A model for DNA-PK DSB recognition within chromatin
Notably, all the cryo-EM data presented here is generated without the use of chemical crosslinkers to stabilise specific structural states, which is unusual in the literature for either nucleosome or NHEJ structures, and we believe indicates the structures we observe have the potential to form in the cell (rather than trapping transient/biologically irrelevant states). From the data presented, we propose a model of DSB recognition by the NHEJ machinery in the context of nucleosomes (Fig. 5). Following initial DSB formation, Ku70/80 is the first protein to recognise the broken DNA ends, binding with high affinity even in the context of nucleosomes. The action of Ku70/80 binding results in the DNA partially "peeling-off" the histones. The position of the break with respect to the nucleosome dyad dictates the specific orientation of Ku70/80 and interactions with H3 and H2A(X) histones. Ku70/80 will always stay at the end of the DNA break and does not freely translocate inwards. DNA-PKcs is then subsequently recruited to form the DNA-PK holoenzyme complex. The binding of DNA-PKcs to nucleosome-bound Ku70/80 causes the vWA domain of Ku80 to open. Opening of the vWA

domain of Ku80 has only previously been observed upon additional NHEJ proteins, such as XLF binding[1,3,25]. Opening of the vWA domain without a known binding partner highlights the importance of DNA-PK binding to nucleosome DNA, potentially "priming" DNA-PK to interact with further NHEJ factors (however, our experiments have not been able to capture complexes formed with additional NHEJ factors). The Ku80-mediated DNA-PK dimer is then stabilised during ATP binding and/or hydrolysis. This dimer is presumably relatively short-lived in the cell, as the addition of non-hydrolysable ATP significantly stabilises this structural form. Remodellers are then likely to be required to remodel the DNA from the nucleosome to allow the recruitment of additional NHEJ proteins and ligation of the broken DNA ends.

In summary, we have captured how Ku70/80 and DNA-PK bind and interact with DSBs when packaged within nucleosomes, allowing us to propose a model for biological NHEJ DNA repair (Fig. 5). We have solved multiple cryo-EM structures tracking the initial DSB recognition and Ku70/80 binding, recruitment of DNA-PKcs and stabilisation of the Ku80-mediated DNA-PK dimer. Our findings reveal unique insights into this critical DNA repair pathway, which will have direct importance for the development of future therapeutics. Specifically understanding the complex mechanism of NHEJ with more physiologically relevant structures (in the context of chromatin) may reveal new unexplored drug pockets, the interplay with chromatin re-modellers in the early stage of repair and unique steps of the mechanism to target. Structural data plays a critical role in rational therapeutic design. Inhibiting NHEJ is beneficial during chemo- and radiotherapy in cancer treatments, and this information paves the way for developing more specific ways to target NHEJ and other repair pathways. Until now, the primary focus for targeting DNA-PKcs has been its ATP-binding site; however, inhibitors directed at this site often lead to significant side effects. The availability of new structural data reveals additional interaction surfaces, such as the interface between the Ku70 C-terminus and the DNA, even when packaged within the nucleosome, which may offer opportunities for developing more selective and less toxic therapeutic agents in the future.

## Methods

### Ku70/80 expression and purification

A pFASTBac-dual expression vector encoding full-length Ku70/80 and Ku70/80-ΔSAP (composed of Ku70 (residues 1–544) + Ku80 full length) with an N-terminal 6 × His-tag on Ku80 were expressed in Sf9 insect cells. Following expression, Ku70/80 was purified by immobilisation onto Ni-NTA beads and anion exchange chromatography. All purification steps were carried out at 4 °C. Cell pellets were resuspended in Lysis Buffer (20 mM Tris, 150 mM KCl, 850 mM NaCl, 5 mM β-mercaptoethanol (BME), 40 mM Imidazole, 1× protease inhibitor tablet, pH 8.0) and sonicated on ice. The resultant cell lysate was incubated, mixing, with MgCl$_2$ (final concentration 10 mM) and 2 μL Benzonase (25 kU of stock) for 20 min and centrifuged at 18,000 × $g$ for 40 min. The supernatant was incubated with Ni-NTA resin (Takara Bio) previously equilibrated with Lysis Buffer for 1 h, and Ku70/80 was eluted with Elution Buffer (20 mM Tris, 150 mM NaCl, 5 mM BME, 300 mM Imidazole, pH 8.0). The elution fractions were analysed on a 10% SDS-PAGE gel, and those containing Ku70/80 were pooled and dialysed against 2 L Buffer A (20 mM Tris, 50 mM NaCl, 50 mM KCl, 5 mM BME, pH 8.0) overnight at 4 °C. The dialysed sample was applied to a 5 mL HiTrap Q column equilibrated in Buffer A, and Ku70/80 was eluted using a linear gradient of Buffer B from 0–40% (20 mM Tris, 150 mM NaCl, 850 mM KCl, 5 mM BME, pH 8.0). Fractions spanning the 280 nm absorbance peak were analysed on a 10% SDS-PAGE gel, the fractions containing Ku70/80 at a high purity were pooled, concentrated and buffer exchanged into cryo-EM Buffer (20 mM HEPES, 200 mM NaCl, 0.5 mM EDTA, 2 mM MgCl$_2$, 5 mM DTT, pH 7.6) using a 50 kDa molecular weight cut-off (MWCO) centrifugal filler (Pierce) and flash-frozen in liquid nitrogen before being stored at −80 °C for further use.

### DNA-PKcs purification

Frozen HeLa Cell Nuclear Extracts were initially dialysed into 5 L Buffer A (20 mM HEPES, 100 mM NaCl, 0.5 mM EDTA, 2 mM MgCl$_2$, 5 mM DTT, 0.2 mM PMSF, 10 complete protease inhibitor cocktail EDTA-free tablets, 10% (vol/vol) glycerol, pH 7.6). DNA-PKcs was isolated using several ion exchange columns: Q Sepharose, HiTrap Heparin, Mono-S and Mono-Q with a NaCl gradient of 0.1–1 M using Buffer B. 4–12% Bis-Tris gels (Invitrogen) were used to confirm the presence and purity of DNA-PKcs. Fractions containing DNA-PKcs were pooled, concentrated, and buffer exchanged into cryo-EM Buffer using a 10 kDa MWCO centrifugal filter (Amicon) and flash-frozen in liquid nitrogen before being stored at −80 °C for further use.

### Histone expression and purification

A vector encoding full-length H2A(X)/H2B/H3/H4 were transformed into Rosetta pLysS cells. A single colony was picked and grown in a 2xYT starter culture before being transferred into 1 L 2xYT media and incubated shaking at 37 °C until the OD$_{600nm}$ reached between 0.6 and 0.8. Protein expression was induced with 0.4 mM IPTG and incubated at 37 °C for 4 h, and cells were harvested.

Cell pellets were resuspended in Histone Wash Buffer (50 mM Tris, 100 mM NaCl, 5 mM DTT, 1× protease inhibitor tablet, pH 7.5) and were sonicated on ice and centrifuged at 30,000 × $g$ for 20 min at 4 °C. The pellet was washed 2× with Histone T-wash Buffer (+2% Triton-100) and then 2× with Histone Wash Buffer before being resuspended in 0.75 mL DMSO and incubated, rolling for 30 min at room temperature. 7.5 mL of Unfolding Buffer (7 M Guanidine Hydrochloride, 20 mM Tris, 10 mM DTT, pH 7.5) was then added and incubated at room temperature, rolling, for 1 h. When the pellet was fully dissolved, Histone Purification Buffer A (7 M Urea, 50 mM Tris, 1 mM EDTA, pH 7.5) was added to bring the total volume to 30 mL before centrifugation. The histone solution was filtered (0.22 μm) and loaded onto a 5 mL HiTrap Q HP column; the flowthrough was collected and subsequently loaded onto a 5 mL HiTrap HP SP column. Histones were eluted by applying a linear gradient of Histone Purification Buffer B (+ 2 M NaCl) from 0–100%. Fractions spanning the 280 nm absorbance peak were analysed on a 15% SDS-PAGE gel. Fractions containing pure histones were pooled and dialysed against 2 L Buffer (5 mM DTT, 0.1% acetic acid, 1× protease inhibitor tablet) and freeze-dried in 2 mg aliquots and stored at −80 °C until required.

### Histone octamer assembly

Individually purified histones H2A(X), H2B, H3 and H4 were resuspended in an Unfolding Buffer (7 M Guanidine hydrochloride, 20 mM sodium acetate, 10 mM DTT, pH 5.2) and mixed in a ratio of 1:1:1:1 at a total protein concentration adjusted to 1 mg per histone in Unfolding Buffer. The histone mix was then dialysed 3× against 2 L Refolding Buffer (2 M NaCl, 10 mM Tris, 1 mM EDTA, 5 mM BME, pH 7.5) at 4 °C, stirring for 2 h, 2 h and overnight. The dialysed sample was injected onto a Superose 6 Increase 10/300 equilibrated with Refolding Buffer. Fractions spanning the main 280 nm absorbance peak were analysed on a 20% SDS-PAGE gel. Fractions containing four histone bands were pooled, concentrated using a 10 kDa molecular weight cut-off centrifugal filter (Amicon) and stored at 4 °C for further use.

### DNA preparation and nucleosome assembly

**147 and 197 bp.** The 147 bp and 197 bp DNA were purchased as ssDNA oligos (IBA Lifesciences) and subsequently amplified by PCR using Taq polymerase (NEB), reaction cycles followed as per the manufacturer's suggestion. The nucleosome DNA and primer sequences can be found in Tables 3 and 4, respectively.

Nucleosome assembly was carried out through the salt gradient dialysis method[6]. The H2A/H2B dimer and H3.1/H4 tetramer at concentrations 1.3 μM and 0.65 μM, respectively, were obtained (EpiMark® Nucleosome Assembly Kit, NEB) and mixed with the appropriate DNA at a concentration of 0.65 μM in Assembly Buffer (20 mM Tris, 1 mM

DTT, 1 mM EDTA, 0.01% Triton ×-100, pH 7.5). The histone/DNA mixture was dialysed against Assembly Buffer (containing 2 M NaCl) for 2 h, followed by Assembly Buffer containing decreasing concentrations of NaCl at 1.5 M, 1 M, 0.6 M, 0.25 M for 2 h each, respectively and finally 0 M NaCl, overnight, at 4 °C, stirring. Nucleosome assembly was confirmed through native PAGE gel analysis and concentrated to a final concentration of 5 µM and stored at 4 °C for further use.

**153 bp.** DNA purification and nucleosome reconstitution followed previously established protocols[28,29]. pUC19 plasmid containing 32 × 153 bp 601 Widom DNA sequence[30] (Supplementary Table 3), was transformed into DH5α cells. A single colony was inoculated into 50 mL LB and grew for 4 h before being inoculated into 750 mL LB, shaking overnight at 37 °C. Cells were harvested and pellets were resuspended in Solution I (25 mM Tris, 50 mM Glucose, 10 mM EDTA, pH 8.0), Solution II (200 mM NaOH, 1% SDS) and Solution III (3 M KOAc, 11% acetic acid) and incubated on ice for 15 min before centrifugation at 11,000 × $g$ for 15 min at 4 °C. The supernatants were filtered using Miracloth (Calbiochem) and 0.52× isopropanol was added. The mix was then incubated for 30 min and centrifuged at 11,000 × $g$ for 30 min at 4 °C. The resulting pellet was then incubated overnight at 37 °C, rotating, with RNAse A (10 mg/mL). To purify the DNA, KCl was added to a final concentration of 2 M, and centrifuged at 11,000 × $g$ for 35 min at 4 °C. The supernatant was filtered (0.22 µm) and injected onto a Sepharose 6 column equilibrated with TES2000 (50 mM Tris, 2 M KCl, 1 mM EDTA, pH 8.0). Fractions spanning the 260 nm absorbance peak were analysed on a 2% agarose gel stained using Ethidium Bromide. The fractions containing plasmid DNA were pooled, and 0.5× isopropanol was added, and centrifuged at 22,000 × $g$ for 20 min at 4 °C. The pellet was resuspended in TE 10,0.1 (10 mM Tris, 0.1 mM EDTA, pH 8.0) and incubated at 50 °C until the DNA was fully dissolved. The plasmid DNA was digested with EcoRV overnight at 37 °C. To precipitate the 153 bp DNA fragment of interest, PEG6000 (40%) with 4 M NaCl was added to a final concentration of 8% and 0.5 M NaCl, incubated on ice for 30 min and centrifuged at 11,000 × $g$ for 30 min at 4 °C. Samples of the pellet and supernatant were run on a 2% agarose gel to confirm the presence of the 153 bp DNA fragment. The pellet was resuspended in TE 10,0.1 and stored at 4 °C until further use.

153 bp DNA was added to a final concentration of 5 µM to varying ratios of the histone octamer, to find the optimal loading of the octamer onto DNA to minimise free DNA in the sample. Ratios 0.6:1.0 to 1.0:1:0 (histone octamer: DNA) were prepared. The assembled samples were placed into dialysis chambers as described in ref. 31 and subject to stepwise dialysis from the Reconstitution Buffer (1.4 M KCl, 10 mM Tris, 0.1 mM EDTA, pH 7.5) into the End Buffer (10 mM KCl, 10 mM Tris, 0.1 mM EDTA, pH 7.5) at 4 °C, stirring. The dialysed samples were removed from the dialysis chambers and centrifuged at 20,000 × $g$ for 10 min at 4 °C. To confirm nucleosome assembly, the samples were run on a 5% acrylamide native gel, against a control of 153 bp free DNA. The nucleosome sample at an equimolar ratio was stored at 4 °C and used in subsequent experiments.

**EMSA studies of nucleosomes with Ku70/80 or Ku70/80-ΔSAP.** For EMSA experiments, different nucleosome to Ku70/80 ratios were attempted to establish which ratio resulted in the best binding of Ku70/80. The nucleosome concentration was kept constant at 50 nM, and Ku70/80 was added in increasing ratios, after assembly samples were incubated on ice for 30 min and run on a 4–12% TBE gel (Invitrogen) for 45 min at 150 Volts with 1× TBE Running Buffer at 4 °C. The best results were obtained when nucleosomes were mixed with Ku70/80 in a 1:4 ratio (nucleosome: Ku70/80). EMSA gels were stained using SybrGOLD (Invitrogen) and imaged under UV light to visualise DNA.

**Purification of the human Ku70 SAP domain in isolation**
A gene fragment (GenScript) coding for the human Ku70 SAP domain was subcloned in frame between the SfoI−XhoI sites in pSF2285[32,33] to generate pMM3046 (Ku70 SAP residues 539–609) for inducible expression in bacteria as a HIS14-SUMObd-Ku70 SAP fusion (the SUMObd moiety is from *Brachypodium distachyon*. In addition, constructs for producing mutant Ku70 SAP domains, pMM3047 (K596A) and pMM3050 (T577E), were similarly generated. All plasmids were verified by DNA sequencing and are available upon request.

The SUMObd-Ku70 SAP fusion plasmids were transformed into bacterial Rosetta 2(DE3)pLysS cells (Novagen), which were grown in standard LB medium (2 L scale) supplemented with kanamycin (25 mg/mL) and chloramphenicol (34 mg/mL) at 37 °C until OD$_{600nm}$ reached 0.6. The temperature of the culture was then dropped to 15 °C, and expression was induced by the addition of IPTG (0.25 mM final concentration) and further incubation for 16 h. Cells were collected by centrifugation, resuspended in 10 mL of PBS supplemented with 10 mM EDTA, and stored at −20 °C. The cell paste was thawed and resuspended with one volume of 1.6 M NaCl, 40 mM Tris-HCl, pH 7.5, 2 mM EDTA, 2 mM DTT, 20 mM imidazole, 0.2% Triton ×-100, 20% glycerol, supplemented with protease inhibitors (Pierce). The lysate was treated by sonication and clarified by centrifugation at 20,000 × $g$. The cleared lysate (~ 50 ml) was injected through a 5 mL HisTrap excel column (Cytiva) equilibrated with 0.8 M NaCl, 20 mM Tris-HCl pH 7.5, 1 mM EDTA, 1 mM DTT, 10 mM imidazole, 10% glycerol, washed with 10 column volumes of the same buffer, and the fusion protein was eluted by increasing the imidazole concentration to 300 mM. The eluate was dialysed at 4 °C for 16 h against 150 mM KCl, 20 mM Tris-HCl, pH 7.5, 1 mM EDTA, 1 mM DTT, 10% glycerol, supplemented with His14-SENP1$_{bd}$ to cleave the SUMO moiety. The dialysate was injected through a 5 mL HisTrap excel column equilibrated with 150 mM KCl, 20 mM Tris-HCl, pH 7.5, 1 mM EDTA, 1 mM DTT, and 10% glycerol to recover the untagged Ku70 domain in the flowthrough. Yields for a 2 L culture were 40 mL at 10 µM, which could readily be concentrated to around 200 µM using 3 kDa MWCO ultrafiltration devices (PALL). Proteins in 150 mM KCl, 20 mM Tris-HCl, pH 7.5, 1 mM EDTA, 1 mM DTT, 10% glycerol were snap frozen in liquid N$_2$ and stored at −80 °C.

**EMSA with the human KU70 SAP domain in isolation**
Binding reactions (10 mL) containing the indicated concentrations of Ku70 SAP protein and 19.3 nM of a 400 bp DNA fragment (generated by PCR using primers 5′-GAGTTTTATCGCTTCCATGAC and 5′-ACTT-GACTCATGATTTCTTACC, with PhiX174 DNA (NEB) as the template) were prepared in 105 mM KCl, 3 mM Tris-HCl (pH 7.5), 0.1 mM EDTA, 0.1 mM DTT, and 4% glycerol. Reactions were incubated for 15 min at room temperature before fractionation on a 0.8% agarose gel (10 cm) in 0.5× TBE buffer, run at 50 Volts for 2 h. Following electrophoresis, gels were stained with GelRed (Biotium) and the fluorescent signal was detected with a ChemiDoc MP imaging system (BioRad).

For assays containing the 153 bp DNA substrate or 153 bp nucleosome, binding reactions were prepared as above, using 50 nM of either free DNA or nucleosome substrate in 46 mM KCl, 3 mM Tris-HCl (pH 7.5), 0.1 mM EDTA, and 3% glycerol. Reactions were incubated and resolved as described above on a 1.5% agarose gel run at 80 V for 120 min at 4 °C. Gels were stained with GelRed and imaged as described.

**Preparation of cryo-EM samples**
**Ku70/80 bound to 147, 153 H2A(X) and 197 bp nucleosome:.** 3.5 µM of the appropriate nucleosome was mixed with 14 µM of Ku70/80, respectively (1:4 ratio) and kept on ice for 30 min. Prior to freezing grids, 3-[(3-Cholamidopropyl)dimethylammonio]-2-hydroxy-1-propanesulfonate (CHAPSO) was added to a final concentration of 8 mM to help circumvent the preferred orientation of particles within vitreous ice. 2.5 µl was then applied to a glow-discharged (30 mA, 30 s) holey carbon grid (Quantifoil 1.2/1.3, 300 mesh) and plunge-frozen into liquid ethane using the FEI Vitrobot system (ThermoFisher Scientific) with the chamber kept at 4 °C and 95% humidity.

**DNA-PK bound to 153 bp H2AX nucleosome:.** 3 µM of the appropriate nucleosome was mixed with 12 µM of Ku70/80 and 6 µM DNA-PKcs, respectively (1:4:2 ratio) and kept on ice for 30 min. CHAPSO was added to a final concentration of 8 mM, and 2.5 µl was then applied to a glow-discharged (30 mA, 30 s) holey carbon grid (Quantifoil 1.2/1.3, 300 mesh) and plunge-frozen into liquid ethane using the FEI Vitrobot system (ThermoFisher Scientific) with the chamber kept at 4 °C and 95% humidity.

**DNA-PK bound to 153 bp H2AX nucleosome with AMP-PNP:.** The complex was assembled as described above, and AMP-PNP was then added to a final concentration of 1 mM and incubated on ice for 1 h before the addition of CHAPSO and grid freezing.

**Cryo-EM data collection:.** Cryo-EM data for the Ku70/80-147 bp nucleosome complex were collected at the Department of Biochemistry, University of Cambridge, on a Krios electron microscope operated at 300 kV using a K3 detector (Gatan Inc.) with EPU data acquisition software (ThermoFisher Scientific). The Ku70/80-197 bp nucleosome complex was collected at the Department of Materials Science, University of Cambridge, on a Krios electron microscope operated at 300 kV using a Falcon III detector with EPU data acquisition software (ThermoFisher Scientific).

Cryo-EM data of all other data presented were collected at The Midlands Regional Cryo-EM facility, University of Leicester, on a Krios electron microscope operated at 300 kV using a K3 detector (Gatan Inc.) with EPU data acquisition software (ThermoFisher Scientific). All data collection parameters can be found in (Supplementary Tables 1 and 2).

**Data processing:.** For the Ku70/80-147 and -197 bp nucleosome complexes, data were processed using Warp[34] and CryoSPARC[35]. In short, motion correction, CTF correction and particle picking were performed using Warp, and particles were extracted in BoxNet2Mask using a box size of 380 and 370 for the 147 and 197 bp nucleosome complexes, respectively. All other data presented were solely processed using CryoSPARC.

The extracted particles were processed using different workflows in cryoSPARC (Supplementary Figs. 3 and 18). In summary, particles were selected and optimised through iterative rounds of 2D classification, ab-initio reconstruction and heterogeneous refinement to separate the Ku- or DNA-PK-nucleosome complexes from free nucleosomes and proteins. The best maps were then further refined using homogenous refinement and finally non-uniform refinement[36]. Local refinements were also completed to improve the density in localised areas. For the obtained final reconstructions, the overall resolution (Å) (displayed in Supplementary Tables 1 and 2) was calculated by Fourier shell correlation at 0.143 cut-off.

**Building and refinement of cryo-EM models:.** For the Ku70/80 bound to 147, 153 (H2A) and 197 bp nucleosome, the model of Ku70/80 (PDB: 1JEY) and for the Ku70/80 bound to 153 bp H2AX nucleosome, the model of Ku70/80 (PDB: 7ZT6) were used as initial templates and rigidly fit into the cryo-EM density maps in UCSF Chimera[37]. For DNA-PK maps, the model of DNA-PK (PDB: 8BH3) was used, with chains other than DNA-PKcs and Ku70/80 manually deleted.

The nucleosome in the 147 bp model (PDB: 7TEEN) and 197 bp (PDB: 6TEM), and for all other models nucleosome (PDB: 6JXD) was used as an initial model. The DNA was manually mutated to the correct sequence, and the DNA extensions were built using the "build" ideal DNA tool in WinCoot[38] and manually fit into the density. The model of the SAP domain (from PDB: 1JEQ) was rigidly fitted in UCSF Chimera, and the linker region was manually built in WinCoot. Once all initial models were generated, Namdinator[39] was used to improve the map to model fit. Following this, each model went through several rounds of

real-space refinement in Phenix[40] and improvement in WinCoot until ready for PDB deposition.

**Cell lines, cell culture and cell engineering:.** U2OS cells (human osteosarcoma cell line from ECACC, Salisbury, UK) were grown in DMEM (Eurobio, France) supplemented with 10% fetal calf serum (Eurobio, France), 125 U/ml penicillin, and 125 µg/ml streptomycin. Cells were maintained at 37 °C in a 5% CO$_2$ humidified incubator. Production of lentiviral particles in HEK-293T cells and transduction of U2OS cells were performed as previously described[41].

U2OS cells expressing the different GFP-tagged Ku70 Linker-SAP constructs were seeded in 35-mm glass-bottom culture dishes (MatTek) 48 h prior to imaging with an Olympus IX73 fluorescence microscope equipped with an Olympus LCAch N 20×/0.40 PhC and a UPlanFL N 40×/0.75 Ph2 objective lenses. Images were taken with an Olympus DP26 camera.

**Expression vectors and DNA manipulations:.** A Ku70 lentiviral expression vector was generated by PCR amplification of human Ku70 cDNA with Ku70-Kpn2-F and Ku70-Bcu-R primers and subsequent insertion between Kpn2I and BcuI restriction sites of the previously described pLV3 vector[25].

Expression vectors for full-length mutant forms of Ku70 were obtained by PCR mutagenesis with the corresponding Ku70-mut-F and Ku70-mut-R oligonucleotides as mutated inner primers and HF-Ku70-Kpn2-F and HF-Ku70-Bcu-R as outer primers, followed by Hot-Fusion insertion of the two overlapping fragments between the Kpn2I and BcuI restriction sites of pLV3[42].

The expression vector for GFP-tagged Ku70 constructs was obtained by replacing the FLAG-Ku70 cDNA from the previously described pLV3-GFP-FLAG-Ku70 vector[43] by a linker cassette (pre-annealed Kpn2-AX-Mlu-F and Kpn2-AX-Mlu-R oligonucleotides) between the Kpn2I and MluI sites, resulting in the pLV3-GFP plasmid. Expression vectors for GFP-tagged Linker, Linker+ or SAP domains were obtained by PCR amplification of the corresponding fragments using full-length Ku70 expression vector as a template with HF-GFP-Ku70-LS-F/HF-Ku70-R554-R, HF-GFP-Ku70-LS-F/HF-Ku70-V557-R or HF-GFP-Ku70-P555-S-F/HF-Ku70-Bcu-R pairs of primers for Linker, Linker+, or SAP domains, respectively. The resulting PCR fragments were inserted by Hot-Fusion between the Kpn2I and BcuI restriction sites of pLV3-GFP.

The expression vectors for mutated Ku70 Linker-SAP constructs were obtained in a similar manner by PCR amplification of the corresponding fragments using full-length Ku70 expression vectors as templates and HF-GFP-Ku70-LS-F and HF-Ku70-Bcu-R as outer primers.

**Reporting summary**
Further information on research design is available in the Nature Portfolio Reporting Summary linked to this article.

## Data availability
All structural data presented are publicly available. Cryo-EM structures and maps are deposited at the PDB and EMDB with accession codes as follows: Ku70/80–147 bp nucleosome (PDB: 9IGW and EMDB: 52860), Ku70/80–153 bp H2A nucleosome (PDB: 9IGX and EMDB: 52861), Ku70/80–153 bp H2AX nucleosome with Ku70 SAP (PDB: 9Q80 and EMDB: 52879), Ku70/80–153 bp H2AX nucleosome no SAP (PDB: 9Q8X and EMDB: 52912), DNA-PK-153 bp H2AX model 1 (PDB: 9Q9F and EMDB: 52958), DNA-PK-153 bp H2AX model 2 (PDB: 9QRC and EMDB: 53025), DNA-PK dimer-153 bp H2AX (PDB: 9QCS and EMDB: 53026) and DNA-PK dimer-153 bp H2AX with ATPγS (PDB: 9QMS and EMDB: 53237). Source data are provided with this paper.

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

## Acknowledgements

We thank Dr Christos Savva, Dr Emma Hesketh, Dr Claudia Lancey and Dr TJ Ragan from the Midlands Regional Cryo-EM Facility at the Leicester Institute of Structural and Chemical Biology (LISCB) and major funding from MRC (MC_PC_17136) for help with grid preparation, screening, data collections and processing support. We also thank EpiCypher for

carrying out experiments to determine Ku70/80 binding to histone peptides. We thank the Lister Institute of Preventative Medicine Prize for support of this research. We would also like to thank the Medical Research Council for the standard research grant (MR/X00029X/1). P.F., S.B. and P.C. were supported by the French National Research Agency (ANR-20-CE11-0026). P.C. is a scientist from INSERM. M.M. thanks Grant AAPG2023–PRC–Project XXL from the French National Research Agency.

## Author contributions

C.H. over-expressed and purified NHEJ proteins and prepared nucleosomes (except the 147 and 197 bp constructs), collected cryo-EM data, processed, modelled the structures and made figures. P.F. carried out functional experiments in cells and data analysis. A.K.S. started the project, over-expressing and purifying NHEJ proteins and assembling 147 and 197 bp nucleosomes and collecting cryo-EM data. A. P. Purified Ku70 SAP domain and carried out EMSAs. S.W.H. processed cryo-EM data, analysed structures and edited the paper. H.A modelled and analysed cryo-EM structures. M.B. constructed original nucleosome constructs. T.M.D.O. helped with cryo-EM data collections and advice. A.T. assisted in nucleosome reconstitution. S.Z. prepared NHEJ proteins. D.Y.C. helped with cryo-EM data collections and analysis. S.B. supervised M.B. with initial nucleosome construct design and assembly. K.M. analysed structures and edited the paper. V.R. Over-expressed and purified Ku70/80 mutants. J.B.C. Directed over-expression and purification of Ku70/80 mutants. M.M. carried out EMSA studies to investigate the SAP domain binding DNA. S.Br. and P.C. guided functional experiments, helped analyse the data and provided ideas. T.L.B. oversaw the start of the project and advised on structural assemblies. T.S. helped with nucleosome design and manuscript editing. A.K.C. directed the study, led the experimental design and wrote the manuscript.

## Competing interests

T.M.D.O. is employed by AstraZeneca.
