## [Transparent Peer Review file · Nature Communications]

Cryo-EM structures of NHEJ assemblies with nucleosomes

Corresponding Author: Dr Amanda Chaplin

Version 0:

Reviewer comments:

Reviewer #1

(Remarks to the Author)

This manuscript by Hall et al., determines the structural basis of the early stages of NHEJ in the context of chromatin. This is accomplished by determining a series of cryo-EM structures that show how the Ku70/80 heterodimer initially engages the terminal ends of a nucleosome during DSB recognition, and how the Ku70/80/DNA-PKc complex assembles on the “broken” nucleosomal DNA ends. These structural insights are timely and substantially increase our understanding for how DNA repair is initiated in chromatin. The work will undoubtedly be well-received in the DNA repair field. However, I do have several comments/concerns related to the quality of the structural models, the structural interpretations, and the cellular assays probing the Ku70 SAP-Linker domain function that the authors should consider and address prior to publication in Nature Communications.

1. The maps are generally not of high enough quality to model the side chain residues within Ku70/80 and Ku70/80/DNA-PKc, and in some cases the NHEJ machinery is modeled in places where there is clearly minimal interpretable cryo-EM density. I strongly suggest the authors go back through these models and prune side chains and remove regions Ku70/80 and Ku70/80/DNA-PKc that do not have interpretable cryo-EM density. This will also likely improve the elevated clash scores for many of these structures.
2. The results section of the manuscript is quite difficult to get through. Part of the issue is the sheer number of different structures the authors walk through, but in most cases this is done in a single sentence or two. For example, the description of the Ku70/80 structure bound to the 197bp WT nucleosome simply states Ku70/80 binds to the end of the linker DNA. Does it thread through the Ku70/80 heterodimer? How does it compare to the mode of Ku70/80 binding to the 147 bp and 153 bp WT nucleosomes? Providing more than a surface level description of many of these structures would likely improve the readability of the manuscript and allow the readers to better track the major take-home message from each of the structures.
3. The authors show quite nicely through complex formation assays that Ku70/80 binds to nucleosomes with varying lengths of linker DNA and to nucleosomes containing histone variants associated with the DNA damage response (e.g., H2Ax). However, the lack of quantitative binding affinities reported within the manuscript makes it very difficult to interpret whether Ku70/80 has any preference for nucleosomes with a specific linker DNA length or nucleosomes with variants. This also makes interpreting the Ku70/80 cryo-EM data difficult, especially given that differences in Ku70/80 binding mode were observed for the different nucleosome substrates (e.g., the SAP domain binding to the acidic patch in H2Ax nucleosomes but not WT nucleosomes). Additional quantitative binding assays would significantly strengthen the manuscript and allow for more robust interpretation of the structural findings. Alternatively, the authors should provide at least some commentary on whether they believe Ku70/80 has some preference for linker DNA of different lengths and/or the histone H2A variants.
4. The putative interaction between the Ku70/80 complex and histone tails in the 147bp WT nucleosome, 153bp WT nucleosome, and 153bp H2Ax nucleosome structures is an intriguing finding. However, it's unclear whether these interactions are important for Ku70/80 binding to the nucleosome, or simply the result of Ku70/80 binding to the nucleosomal DNA ends placing it in proximity to the H3 and C-terminal H2A tail. In addition, it was difficult to track why these interactions were seen in some of the structures but not others? Without higher resolution structural information where the interaction can be seen in detail, or supporting biochemical assays that show this interaction is important for Ku70/80 complex binding to nucleosomes, the authors should use caution when discussing the histone tail interactions throughout the manuscript.

5. The observation that entry/exit site DNA is partially unwrapped when bound by Ku70/80 in the 147bp and 153bp nucleosomes could also be explained by spontaneous unwrapping of the nucleosomal DNA that allows Ku70/80 to capture the nucleosomal DNA ends (see PMID: 1525856), rather than Ku70/80 physically “peeling” the nucleosomal DNA off the histone octamer after binding. This alternative interpretation should at least be mentioned in the manuscript as well as the discussion.

6. The authors note that they did not observe two Ku70/80 heterodimers bound to the same DNA end in all of the different nucleosomes structures they determined with varying linker DNA length (Lines 131-142). The cryo-EM processing workflow in Extended Fig. 2 lacks any of the intermediate maps generated during processing, which makes it impossible to evaluate a statement such as this. While I appreciate the attempts to consolidate as much of the cryo-EM processing into single figures, Extended Fig. 2 (and to some extent Extended Fig. 6) are overly complicated making it difficult to track the processing workflow for each individual dataset.

7. Can the authors comment on why the Ku70 SAP-linker only interacts with the acidic patch and nucleosomal DNA in the H2Ax nucleosome complex but not the WT nucleosome complex? This intriguing finding was given minimal explanation.

8. The experiments testing the importance of the SAP domain and linker region of Ku70 in U2OS cells are problematic. Some of these issues and points requiring clarification are below:

- Why did the authors choose to study the Ku70 SAP and linker domain without the rest of the Ku70 protein?
- It appears that the SAP domain and/or linker mutants were never tested for their ability to bind nucleosomes in vitro prior to moving into the cellular experiments. This makes interpreting these experiments extremely difficult, as it's unclear whether these mutants actually disrupt the SAP-linker domain interaction with the nucleosome that was observed in the structures.
- Were these cells treated with DNA damaging agents to generate DSBs to recruit Ku70/80 to chromatin? If not, is it possible that some of the mutants that show null results are because the SAP-linker interface is important for DSB recognition in chromatin rather than just general chromatin binding?
- Did the authors attempt to quantify any of these experiments?
- I was unable to find the methods section for this set of experiments, which should be addressed.

9. The structures of Ku70/80/DNA-PKc (non-activated) bound to the H2Ax nucleosome are quite remarkable. The authors did an excellent job walking through the different conformational states related to Ku80 vWa opening/closing. However, they fail to describe how the Ku70/80/DNA-PKc complex bound to the H2Ax nucleosome differs from the earlier described Ku70/80 complex bound to H2Ax nucleosome. Does the overall conformation of the nucleosome change between these structures? Does Ku70/80 bind to the nucleosome in a similar manner in these structures regardless of the presence of DNA-PKc? What happens to the SAP-linker domain interaction with the acidic patch and the nucleosomal DNA? My own initial comparisons of these structures suggest quite a few interesting differences that were either overlooked or ignored. This type of structural analysis is quite important given that the novelty of this work is rooted in understanding how the NHEJ machinery functions within a chromatin context (on a nucleosome substrate).

10. A similar comparison of the Ku70/80/DNA-PKc bound to the H2Ax nucleosome (non-activated states) and the Ku70/80/DNA-PKc bound to the H2Ax nucleosome (activated states) should be made (see preceding comment).

11. The section of the discussion on drug development is quite superficial. What unique insights were gleaned from these structures that will enhance the development of therapeutics?

Minor Comments

1. Several of the structural models provided with the manuscript have components that vary in chain ID from model to model, which made it quite difficult to compare/contrast the structures (e.g., compare Ku70/80 bound to 153bp H2Ax nucleosome and Ku70/80/DNA-PKc bound to 153bp H2Ax nucleosome-model 1). The consensus used in the field is that Chain A/E (Histone H3), Chain B/F (Histone H4), Chain C/G (Histone H2A), Chain D/H (Histone H2B), Chain I/J (Nucleosomal DNA). I would suggest the authors renumber the chains within the models for uniformity between their own structures, but also for uniformity with virtually all other nucleosome structures in the PDB.

2. The manuscript contains numerous grammatical errors and labeling errors. Some, but not all, are highlighted below.

3. Lines 91-92: A figure showing a structural comparison of Ku70/80 engaged with the nucleosome and the previously determined structures of Ku70/80 bound to DNA would be useful.

4. Extended Data Fig. 2: Particle numbers for 147bp DNA have commas in the wrong place.

5. Lines 99-100: This sentence lacks clarity.

6. Lines 140-141: I believe “permissive” should be “not permissive”.

7. Lines 167-171: This section should be edited for clarity. The resolution of this 153bp nucleosome bound by two Ku70/80 should be given in the sentence it was described in rather than the sentence after.

8. Lines 179-180: This sentence lacks clarity.

9. Line 200: I believe "K543" should be "R543" based on Figure 2b.
10. Line 256-258: This statement requires a citation.
11. Lines 358-360: I would caution the authors in describing this as "unwinding the DNA."

Reviewer #2

(Remarks to the Author)

The premise of this study is both interesting and promising. DNA breaks in eukaryotes occur within chromatin, and thus DNA double-strand breaks (DSBs) are surrounded by nucleosomes. However, no studies to date have shown how the repair initiators Ku70/80 and DNA-PKcs interact with nucleosomes.

In this manuscript, the authors report structures of Ku70/80 bound to the end(s) of 147 bp (Widom 601 sequence), 153 bp and 197 bp DNA on a nucleosome. In these constructs, however, the nucleosome is positioned centrally with both DNA ends being symmetrically equal in length. These configurations do not reflect the typical scenario in which only one end is accessible and free for binding by Ku70/80, or by both Ku70/80 and DNA-PKcs. The authors determined cryoEM structures of these three nucleosomes bound by Ku70/80. As expected, 147 bp and 153 bp nucleosomes do not present enough free DNA for Ku70/80 and DNA-PKcs to bind, so DNA ends are peeled off from the histones. In the case of the 147 bp nucleosome, only one end is peeled off, resulting in partial binding by Ku70/80.

In addition, the authors analyzed potential effects of the histone variant H2AX, which replaces H2A on damaged DNA, on Ku70/80 binding, as well as the influence of DNA-PKcs and ATP on the combined binding of Ku70/80 and DNA-PKcs to these nucleosomes. These findings raise more questions than answers (see comments below). Due to the lack of structural knowledge regarding DSB repair in the context of nucleosomes, the results presented in this manuscript are novel and warrant consideration for publication. However, several issues related to DNA substrate design and data interpretation should be addressed first. A scientist should recognize the limitations of each experiment and interpret the results based on experimental data and established scientific knowledge.

1. The three nucleosome DNAs used in this study as substrates for DSB repair are unrealistic. First, in a cellular context, two DNA breaks surrounding a nucleosome would likely lead to the loss of the nucleosome rather than its repair. Second, it is highly improbable that a DSB can occur so close to or even within a nucleosome, such as the 147 bp one, because a histone octamer would either slide off or move inward from the broken end, even if the DSB occurs immediately adjacent to a nucleosome, regardless of the presence of nucleosome remodelers. Third, the 601 Widom sequence binds tightly to a histone octamer and is not prone to sliding in either direction. This is evident from the structures of the 153 bp nucleosome-Ku70/80 complex, where only a limited number of DNA base pairs were "peeled" off at each end, and neither end results in optimal binding by Ku70/80.
2. The authors observed that one end of the DNA in the 147-bp nucleosome is associated with Ku70/80 when a 4-fold excess of Ku70/80 is mixed with the nucleosome. Is the DNA end bound by Ku70/80 unique, or is it a mixture of both DNA ends? Did the histone core translocate along the DNA?
3. How many base pairs are associated with Ku70/80 in the 147-bp nucleosome? Typically, a minimum of 14 bp are covered by one Ku70/80 molecule (PMID: 11493912). In the case of the 147 bp Widom nucleosome, approximately 140 bps are involved in histone binding. Peeling off any stretch of DNA to allow Ku70/80 to bind would be energetically costly. Ku70/80 binds free DNA ends with a dissociation constant of ~ 1 nM; however, the K_d of Ku70/80 when associated with nucleosomes appears to range from 50 to 100 nM, indicating it is 50-100 times weaker. Could such weak binding occur in vivo?
4. The authors asserted multiple times that Ku70/80 is the first to arrive at broken DNA ends. What evidence do they have to support this statement? Recent measurements in living cells indicate that Ku70/80 and DNA-PKcs arrive at broken DNA ends simultaneously (PMID: 39578493). Furthermore, as quoted by the authors in reference #23, DNA-PKcs limits excessive loading of Ku70/80. If Ku70/80 arrives before DNA-PKcs, how does DNA-PKcs prevent the loading of multiple Ku70/80 onto a DNA end?
5. Human Ku70/80 typically slides along free DNA and does not remain at the DNA end in the absence of DNA-PKcs. Therefore, the observation of Ku70/80 remaining at the DNA ends on the 197-bp nucleosome is unusual. Is there steric hindrance preventing Ku70/80 from sliding toward the nucleosome? For studying Ku70/80 and DNA-PKcs binding, an ideal nucleosome may consist of 175 bp, leaving one free DNA end of 30 bp.
6. The cell data reporting SAP binding to chromatin is promising. However, it does not differentiate whether SAP binds to the acidic patch of the histone or the DNA, nor does it specify whether it interacts with H2A or H2AX. As a result, the implications of SAP binding to the acidic patch in the 153-bp nucleosome containing H2AX remain uncertain. Given the spatial constraint, it is possible that SAP docking on the histone acidic patch, which is common to both H2A and H2AX, could be coincidental.
7. As the acidic patch exists in both H2A and H2AX, have the authors measured the binding affinity of the SAP domain of Ku70 with H2A/H2B compared to H2AX/H2B? Additionally, how does SAP binding to the acidic patch compare to that of known binder of H2A, such as Dot1L, SAGA DUB or COMPASS?
8. In the presence of DNA-PKcs, how many base pairs of the 153 bp nucleosome are bound by DNA-PKcs? A superimposition of the structure of DNA-PK bound to free DNA with that of the 153-bp nucleosome would be helpful. Only when 15 bp are bound to DNA-PKcs is the kinase activated. Have the authors measured the DNA-PKcs kinase activity in the presence of 147- or 153-nucleosome?
9. Given that the DNA end is blunt and open, did the authors observe specific interactions between DNA-PKcs and the DNA end, such as the DEB (DNA end blocking) helix of DNA-PKcs between the DNA strands? DEB binding to the DNA end may slow down but not completely inhibit DNA-PKcs kinase activity.
10. Given the extremely short DNA length available for DNA-PKcs binding and the lack of evidence supporting DNA-PKcs activation, the conclusions regarding the roles of ATP and AMP-PNP in promoting DNA-PK dimerization and the "activated

state" of DNA-PKcs are speculative, if not entirely unfounded.

Reviewer #3

(Remarks to the Author)

The authors present cryo-electron microscopy structures of the human Ku70/80 heterodimer and DNA-PK holoenzyme bound to nucleosomes with varying DNA lengths, to investigate how non-homologous end joining machinery recognizes DNA double-strand breaks in chromatin. They show that Ku70/80 binding can unwrap nucleosomal DNA and find that the Ku70 SAP domain binds nucleosomal DNA in the Ku complex. Further interactions are found at the histone acidic patch and involve the linker region between the Ku80 core domain and the SAP domain. Additionally, they reveal a conformational change in Ku80 upon DNA-PK binding and demonstrate that non-hydrolysable ATP promotes DNA end synapsis. Altogether, these findings advance our understanding of the binding of Ku to chromatinized DNA and provide structural insights into DSB repair by NHEJ. All in all, the paper is well written and the structural analysis generally solid. The paper covers structural biology up to some cell based assays. The area of research is of interest to both basic and biomedical sciences. In general, I believe the manuscript is of interest to the readers of Nature Communications. However, there are some parts where I believe the evidence needs to be solidified before I can recommend publication.

Major points:

1) Line 171/Line 179-180: Is the appearance of the extra density and the proposed stabilisation of the flexible tail specific to H2AX? This is somewhat surprising since the acidic patch is also present in canonical histones. Are there any differences between H2AX and H2A at contact points between Ku and the nucleosome? Or can altered DNA unwrapping account for the observed difference, and if so, would that be specific to H2AX? To validate the differences in binding, perhaps you could perform a competition experiment to see whether H2AX nucleosomes are bound better than H2A nucleosomes?

2) What is the origin of the specific model shown in Figure 2E? The authors do not really show detailed density for that part and even state that they could not confidently identify specific histone contacts solely from cryo-EM maps. How can they confidently model the linker? Does AlphaFold predict this interaction? I am not yet convinced on the basis of the provided data for the physiological relevance of the SAP and linker nucleosome binding. Since this is a main part and conceptual advance in this manuscript, I suggest that the authors better characterise this interaction *in vitro*, e.g. using a construct that contains only linker and SAP domains and demonstrate nucleosome and acidic patch binding and work out any differences between H2AX and H2A nucleosomes.

Minor and editorial points:

Line 53: Perhaps define DNA-PK as DNA-dependent protein kinase first before abbreviating it.

Line 161: Please indicate at which position DNA-PKcs phosphorylates H2AX.

Figure 2 A and C : These panels overlap with gel labels ("Nucleosomes") covered by Figure 2C.

Figure 2 and general: Consider standardising the use of capital vs. lowercase letters.

Extended Figure 2 : The local refinement depictions appear somewhat blurry (also in Extended Figure 5). In general, the processing scheme images are not very sharp. Perhaps this is due to the downsampled review files, but if not, please consider improving their resolution and contrast.

Version 1:

Reviewer comments:

Reviewer #1

(Remarks to the Author)

The authors have thoroughly addressed my initial comments/concerns, with one exception. I still believe that the cellular experiments carried out to define the functional importance of the SAP-linker in chromatin binding are underdeveloped and difficult to interpret, as was also noted by the other reviewers. Given the inability to characterize any of the SAP-linker mutants *in vitro*, the use of a truncated SAP-linker for the cellular experiments, and the absence of DSBs in these cellular experiments, it's unclear how the findings connect back to the role of the SAP-linker in Ku70/Ku80 DSB recognition in chromatin that the authors thoroughly describe in their structural analysis. I do appreciate the technical difficulties the authors noted for these experiments, and so I feel it best to default to the editor to decide if this was sufficiently addressed and/or whether these experiments in their current state add any value to the manuscript. I must stress that the rest of this manuscript represents a significant advance in the DNA repair field, and I do feel the manuscript is otherwise suitable for publication in Nature Communications. I applaud the authors for taking on this challenging project.

Minor Comments

1. The gel in Supplementary Figure 17 appears to be mislabeled. I believe the right side of the gel should be +Ku70/80- Δ SAP.

Reviewer #2

(Remarks to the Author)

The revised manuscript includes changes in wording, but it lacks any new experiments necessary to assess the significance of the structural observations made with the Ku70/80 and nucleosome complexes. ED Fig. 17 does not present the binding affinities of full-length Ku70/80 and the Δ SAP variant to free DNA; instead, it demonstrates the loading of multiple Ku70/80 molecules onto a 153-bp oligo DNA. Additionally, there are no measurements of the binding affinity of Ku70/80 to nucleosomes (147 bp or 153 bp) as requested in the first round of review.

The following measurements are necessary for the reconsideration of the manuscript:

- (a) Kd of Ku70/80-FL and Ku70/80- Δ SAP to free DNA oligonucleotides, e.g., 20 bp.
- (b) Kd of Ku70/80-FL and Ku70/80- Δ SAP to 147-bp and 153-bp nucleosomes.

To avoid loading of multiple Ku70/80 molecules onto one DNA duplex, Kd can be measured with binding partners at a constant molar ratio of 1:1.

Reviewer #3

(Remarks to the Author)

The authors satisfactorily addressed my points. I appreciate efforts to better understand SAP domain nucleosome binding and the manuscript is revised appropriately. I am happy with the current version and can support publication.

Version 2:

Reviewer comments:

Reviewer #1

(Remarks to the Author)

The authors have addressed my remaining concerns.

Reviewer #2

(Remarks to the Author)

The EMSA data published or presented in this manuscript support the conclusion that the SAP domain binds to DNA; however, it does not provide evidence for binding to histones. Furthermore, the mutant analysis reinforces the importance of DNA binding by the SAP domain, but it does not address whether SAP binds to chromatin proteins, contrary to the implications made by the authors.

The manuscript lacks data to substantiate the claim that Ku70/80 binds to nucleosomes beyond its interaction with DNA.

Response to Reviews

Thank you for arranging the review of our manuscript for Nature Communications. We thank all the reviewers for their careful consideration of the manuscript and were pleased to read that they agreed the results increase our understanding of the NHEJ machinery and its interaction with chromatin. We have now addressed the reviewer's comments in our significantly revised submission and have detailed our responses below. We have included additional data analysis of all the structures with multiple new supplementary figures, redeposited the improved PDBs and added additional EMSA studies to analyse the Ku70 SAP domain importance.

We look forward to your response to our changes.

Your Sincerely,

Amanda Chaplin and co-authors

REVIEWER COMMENTS

Reviewer #1:

This manuscript by Hall et al., determines the structural basis of the early stages of NHEJ in the context of chromatin. This is accomplished by determining a series of cryo-EM structures that show how the Ku70/80 heterodimer initially engages the terminal ends of a nucleosome during DSB recognition, and how the Ku70/80/DNA-PKc complex assembles on the "broken" nucleosomal DNA ends. These structural insights are timely and substantially increase our understanding for how DNA repair is initiated in chromatin. The work will undoubtedly be well-received in the DNA repair field. However, I do have several comments/concerns related to the quality of the structural models, the structural interpretations, and the cellular assays probing the Ku70 SAP-Linker domain function that the authors should consider and address prior to publication in Nature Communications.

We thank the reviewer for mentioning that this work is timely and substantially increases our understanding for how DNA repair is initiated in chromatin. We appreciate the comments and will respond in detail below.

1. The maps are generally not of high enough quality to model the side chain residues within Ku70/80 and Ku70/80/DNA-PKc, and in some cases the NHEJ machinery is modelled in places where there is clearly minimal interpretable cryo-EM density. I strongly suggest the authors go back through these models and prune side chains and remove regions Ku70/80 and Ku70/80/DNA-PKc that do not have interpretable cryo-EM density. This will also likely improve the elevated clash scores for many of these structures.

We appreciate the reviewers' comment, although the cryo-EM density is clear for modelling the individual proteins and DNA, we agree that in some areas of density for the side chain residues is limited. We have now gone back through the data and improve where possible the modelling by removing side chains which have no clear density. We have also modified the

chain IDs to be consistent as mentioned by another reviewer. Hopefully this will enable easier comparisons between the structures.

2. The results section of the manuscript is quite difficult to get through. Part of the issue is the sheer number of different structures the authors walk through, but in most cases this is done in a single sentence or two. For example, the description of the Ku70/80 structure bound to the 197bp WT nucleosome simply states Ku70/80 binds to the end of the linker DNA. Does it thread through the Ku70/80 heterodimer? How does it compare to the mode of Ku70/80 binding to the 147 bp and 153 bp WT nucleosomes? Providing more than a surface level description of many of these structures would likely improve the readability of the manuscript and allow the readers to better track the major take-home message from each of the structures.

We apologise if the results section was too concise, due to the large number of structures we initially felt a concise description of each would be clear and focused. However, we have now expanded the results section and included greater detail, including information mentioned above for the 197bp structure. We have also added an additional supplemental figure (**Supplementary Fig. 7**) showing how Ku70/80 with linear DNA (PDB: 1JEY)¹ compares to Ku70/80 once bound to the nucleosomal DNA. We feel this broadens and expands our description of the structures presented.

3. The authors show quite nicely through complex formation assays that Ku70/80 binds to nucleosomes with varying lengths of linker DNA and to nucleosomes containing histone variants associated with the DNA damage response (e.g., H2Ax). However, the lack of quantitative binding affinities reported within the manuscript makes it very difficult to interpret whether Ku70/80 has any preference for nucleosomes with a specific linker DNA length or nucleosomes with variants. This also makes interpreting the Ku70/80 cryo-EM data difficult, especially given that differences in Ku70/80 binding mode were observed for the different nucleosome substrates (e.g., the SAP domain binding to the acidic patch in H2Ax nucleosomes but not WT nucleosomes). Additional quantitative binding assays would significantly strengthen the manuscript and allow for more robust interpretation of the structural findings. Alternatively, the authors should provide at least some commentary on whether they believe Ku70/80 has some preference for linker DNA of different lengths and/or the histone H2A variants.

We thank the reviewer for this comment and agree quantitative affinities would be ideal. However, we should stress that we have been trying to obtain these over the last few years, and it appears to be more difficult than what one would expect. ITC is very difficult as it is not possible to reconstitute nucleosomes at a high enough concentration. BLI has been tried and unfortunately was not successful. Ku70/80 was immobilised on BLI sensors via the His-tag on the protein, with the nucleosome present in the wells. Loading onto the sensors was successful, however no association or dissociation was detected with nucleosomes. We therefore tried our usual Y-shaped DNA as a positive control (known to have a nM affinity) and still limited association and dissociation was detected with a lot of background signal (DNA binding to reference, unloaded sensor). We have also tried biotinylated Ku70/80 and immobilised onto streptavidin sensors (less background binding), however we still observed no association or dissociation. We now believe that having Ku immobilised is causing steric hindrance of the central DNA binding channel and not allowing binding. We have included these results below to demonstrate everything we have tried in **Figure 1 - below**.

We agree that we should have made it clear in the manuscript that Ku70/80 does not have an observed preference for DNA sequence or length or whether it is in the presence of H2A or H2AX. Although we report density for the Ku70-SAP domain in the H2AX structure, upon analysis in our nucleosome data containing H2A, we do still observe some density for the SAP domain. We have now shown this in an additional supplemental figure (**Supplementary Fig. 14**). This density was however more difficult to resolve than that presented for the H2AX dataset, but this is now described in more detail within the text. We have also now included an additional EMSA to show how when the SAP domain is removed from full-length Ku protein that an increased band shift can be observed, again demonstrating the importance of the SAP domain for DNA binding (**Supplementary Fig. 17**). Expression and purification of the SAP domain in isolation was also carried out, however it was not shown to bind to DNA with a high affinity and therefore was difficult to analyse the effect of the point mutation (**Figure 2 - below**).

Figure 1: Biolayer Interferometry (BLI) and Flow induced dispersion analysis (FIDA) experiments. **A)** Ni-NTA biosensor + His-tagged Ku70/80. Loading of Ni-NTA with His-Ku 200-25nM (+ reference “unloaded” sensor) for 600 seconds. Association into “analyte”, 153 bp H2AX nucleosome 100nM for 300 seconds. No association or dissociation. **B)** Linear free DNA (Y-shape, walker) as a positive control for Ku binding. No association or dissociation between Ku70/80 and nucleosome DNA or linear DNA. Perhaps Ku70/80 being immobilised causes steric hinderance of the Ku70/80 DNA binding channel. **C)** Biotinylated Ku70/80 and immobilised onto streptavidin biosensors. Ku70/80 loading onto streptavidin sensors 500nM → 31.25nM. Analyte (y-shape DNA – Walker et al.,2001) concentration 200nM. No association or dissociation between Ku70/80 and linear DNA. **D-E)** FIDA-Bio experiments. **D)** Ku70/80 full-length protein. **E)** Ku70/80 truncated SAP. **F)** 153bp nucleosome. **G)** Ku70/80 truncated SAP with 153 bp H2AX nucleosome at a ratio of 1:4 (5:20 μM). Averaging technique, not separation technique. Having an excess of Ku is masking any visualisation of a potential complex. However, to stay consistent with the EMSA and cryo-EM results it was assembled complex at 1:4 ratio.

Figure 2: EMSA gels of Ku70 SAP domain. A) WT Ku70 SAP domain binding to DNA compared to K596 SAP mutant binding to 30 bp DNA. B) WT Ku70 SAP domain binding to DNA compared to K596 SAP mutant binding to 153 bp DNA. C) WT Ku70 SAP domain binding to DNA compared to K596 SAP mutant binding to 153 bp DNA with higher ratios and lower DNA concentration.

4. The putative interaction between the Ku70/80 complex and histone tails in the 147bp WT nucleosome, 153bp WT nucleosome, and 153bp H2Ax nucleosome structures is an intriguing finding. However, it's unclear whether these interactions are important for Ku70/80 binding to the nucleosome, or simply the result of Ku70/80 binding to the nucleosomal DNA ends placing it in proximity to the H3 and C-terminal H2A tail. In addition, it was difficult to track why these interactions were seen in some of the structures but not others? Without higher resolution structural information where the interaction can be seen in detail, or supporting biochemical assays that show this interaction is important for Ku70/80 complex binding to nucleosomes, the authors should use caution when discussing the histone tail interactions throughout the manuscript.

We again do thank the reviewer for this comment and recognise the limitations of these observed interactions. We should mention that we have explored direct interactions between the histone tails and Ku70/80 using the company EpiCypher. They explored interactions between 280 biotinylated histone peptides with and without post-translational modifications and their interactions with Ku70/80. However, they were unable to find a direct interaction between the histone tails we observe as interacting with Ku70/80. These interactions may therefore only be present within the context of the nucleosomes. We have now added this information into the manuscript and as the reviewer mentions discussed these possible interactions with caution (**Supplementary Fig. 10**).

5. The observation that entry/exit site DNA is partially unwrapped when bound by Ku70/80 in the 147bp and 153bp nucleosomes could also be explained by spontaneous unwrapping of the nucleosomal DNA that allows Ku70/80 to capture the nucleosomal DNA ends (see PMID: 1525856), rather than Ku70/80 physically “peeling” the nucleosomal DNA off the histone octamer after binding. This alternative interpretation should at least be mentioned in the manuscript as well as the discussion.

This is a very nice alternative that we had not considered and have now included this possibility within the manuscript.

6. The authors note that that they did not observe two Ku70/80 heterodimers bound to the same DNA end in all of the different nucleosomes structures they determined with varying in linker DNA length (Lines 131-142). The cryo-EM processing workflow in Extended Fig. 2 lacks any of the intermediate maps generated during processing, which makes it impossible to evaluate a statement such as this. While I appreciate the attempts to consolidate as much of the cryo-EM processing into single figures, Extended Fig. 2 (and to some extent Extended

Fig. 6) are overly complicated making it difficult to track the processing workflow for each individual dataset.

We appreciate what the reviewer is saying, however, we must say these workflows did take a significant amount of time, in order to condense what would have been 9 additional figures. However, we do agree that they do not include intermediate maps and so we have now included these in additional figures (**Supplementary Figs. 4, 5, 19 and 20**).

7. Can the authors comment on why the Ku70 SAP-linker only interacts with the acidic patch and nucleosomal DNA in the H2Ax nucleosome complex but not the WT nucleosome complex? This intriguing finding was given minimal explanation.

We apologise for the minimal explanation, as mentioned above we believe the SAP and linker can bind when H2A is present but seems to appear stabilised and easily resolved with H2AX. We have now added additional description and detail about this including a figure (**Supplementary Fig. 14**) showing its presence in the H2A dataset.

8. The experiments testing the importance of the SAP domain and linker region of Ku70 in U2OS cells are problematic. Some of these issues and points requiring clarification are below:

- Why did the authors choose to study the Ku70 SAP and linker domain without the rest of the Ku70 protein?

Without DNA-breaking treatment of cells, full-length (FL) Ku heterodimer exhibits a nuclear localization and is mostly extractable from chromatin, except a small RNase-sensitive fraction², with no peculiar mitotic chromatin labelling. So, we chose to limit the analysis to the Ku70 (Linker)-SAP region since its intrinsic chromatin tropism allowed exploration of mutations effects.

- It appears that the SAP domain and/or linker mutants were never tested for their ability to bind nucleosomes *in vitro* prior to moving into the cellular experiments. This makes interpreting these experiments extremely difficult, as it's unclear whether these mutants actually disrupt the SAP-linker domain interaction with the nucleosome that was observed in the structures.

We agree that *in vitro* nucleosomes binding experiments with WT and mutant linker-SAP-linker would help the interpretation of the cellular data. As mentioned above we have now included an additional EMSA gel with and without the SAP domain, showing how the SAP domain can control Ku70/80 loading (**Supplementary Fig. 17**). We have also attempted to purify the SAP domain and point mutation separately, and although the protein purified well, we could not observe any binding to DNA using EMSAs (**Figure 2 - above**). Therefore, it was not possible to carry out any further *in vitro* studies.

- Were these cells treated with DNA damaging agents to generate DSBs to recruit Ku70/80 to chromatin? If not, is it possible that some of the mutants that show null results are because the SAP-linker interface is important for DSB recognition in chromatin rather than just general chromatin binding?

The cells were not treated with DNA breaking agents, so that the mitotic labelling we show is intrinsic and distinct from DSB recognition. We agree that some positions of the linker-SAP-linker domain the mutation of which do not impact on intrinsic binding to chromatin may contribute to full length Ku binding to DSB.

- Did the authors attempt to quantify any of these experiments?

We did not attempt quantification since the effect of mutations were sharp, with either no effect (with the T577A SAP mutant, each mitotic chromatin was labelled as WT SAP control) or full effect (no mitotic chromatin was labelled with either K596A or K596E SAP mutant).

• I was unable to find the methods section for this set of experiments, which should be addressed.

We apologize for this inadvertent omission. Here is the corresponding method paragraph that will be added to the Methods section (Cell engineering and imaging).

“U2OS cells expressing the different GFP-tagged Ku70 Linker-SAP constructs were seeded in 35-mm glass-bottom culture dishes (MatTek) 48h prior to imaging with an Olympus IX73 fluorescence microscope equipped with an Olympus LCAch N 20X/0.40 PhC objective lens. Images were taken with an Olympus DP26 camera.”

9. The structures of Ku70/80/DNA-PKc (non-activated) bound to the H2Ax nucleosome are quite remarkable. The authors did an excellent job walking through the different conformational states related to Ku80 vWa opening/closing. However, they fail to describe how the Ku70/80/DNA-PKc complex bound to the H2Ax nucleosome differs from the earlier described Ku70/80 complex bound to H2Ax nucleosome. Does the overall conformation of the nucleosome change between these structures? Does Ku70/80 bind to the nucleosome in a similar manner in these structures regardless of the presence of DNA-PKc? What happens to the SAP-linker domain interaction with the acidic patch and the nucleosomal DNA? My own initial comparisons of these structures suggest quite a few interesting differences that were either overlooked or ignored. This type of structural analysis is quite important given that the novelty of this work is rooted in understanding how the NHEJ machinery functions within a chromatin context (on a nucleosome substrate).

We thank the reviewer for this comment and agree that a more detailed comparison is necessary. We have now added additional text within the manuscript, highlighted in red and an additional figure **Supplementary Fig. 7**. The reviewer is right there are a few differences that should have been highlighted, particularly the movement of the Ku70-SAP domain, that we have now included.

10. A similar comparison of the Ku70/80/DNA-PKc bound to the H2Ax nucleosome (non-activated states) and the Ku70/80/DNA-PKc bound to the H2Ax nucleosome (activated states) should be made (see preceding comment).

We again thank the reviewer and have now added additional textual comparisons and added an additional **Supplementary Fig. 24**.

11. The section of the discussion on drug development is quite superficial. What unique insights were gleaned from these structures that will enhance the development of therapeutics? We have now added some more specific text on this.

Minor

Comments

1. Several of the structural models provided with the manuscript have components that vary in chain ID from model to model, which made it quite difficult to compare/contrast the

structures (e.g., compare Ku70/80 bound to 153bp H2Ax nucleosome and Ku70/80/DNA-PKc bound to 153bp H2Ax nucleosome-model 1). The consensus used in the field is that Chain A/E (Histone H3), Chain B/F (Histone H4), Chain C/G (Histone H2A), Chain D/H (Histone H2B), Chain I/J (Nucleosomal DNA). I would suggest the authors renumber the chains within the models for uniformity between their own structures, but also for uniformity with virtually all other nucleosome structures in the PDB.

We have now gone through the models and changed the chain IDs so they are now consistent and make for easier comparison.

2. The manuscript contains numerous grammatical errors and labeling errors. Some, but not all, are highlighted below.
3. Lines 91-92: A figure showing a structural comparison of Ku70/80 engaged with the nucleosome and the previously determined structures of Ku70/80 bound to DNA would be useful. We have now included this in **Supplementary Fig. 7** and added additional text.
4. Extended Data Fig. 2: Particle numbers for 147bp DNA have commas in the wrong place. Thank you, this has now been changed.
5. Lines 99-100: This sentence lacks clarity. This sentence has now been improved for clarity.
6. Lines 140-141: I believe “permissive” should be “not permissive”. We have now changed this.
7. Lines 167-171: This section should be edited for clarity. The resolution of this 153bp nucleosome bound by two Ku70/80 should be given in the sentence it was described in rather than the sentence after. This has now been changed.
8. Lines 179-180: This sentence lacks clarity. This has now been deleted and reworded.
9. Line 200: I believe “K543” should be “R543” based on Figure 2b. This has now been changed.
10. Line 256-258: This statement requires a citation. This has now been added.
11. Lines 358-360: I would caution the authors in describing this as “unwinding the DNA.” This has now been changed to remodel.

Reviewer #2

The premise of this study is both interesting and promising. DNA breaks in eukaryotes occur within chromatin, and thus DNA double-strand breaks (DSBs) are surrounded by nucleosomes. However, no studies to date have shown how the repair initiators Ku70/80 and DNA-PKcs interact with nucleosomes.

We thank the reviewer for noting that this study is both interesting and promising.

In this manuscript, the authors report structures of Ku70/80 bound to the end(s) of 147 bp (Widom 601 sequence), 153 bp and 197 bp DNA on a nucleosome. In these constructs, however, the nucleosome is positioned centrally with both DNA ends being symmetrically equal in length. These configurations do not reflect the typical scenario in which only one end is accessible and free for binding by Ku70/80, or by both Ku70/80 and DNA-PKcs. The authors determined cryoEM structures of these three nucleosomes bound by Ku70/80. As expected, 147 bp and 153 bp nucleosomes do not present enough free DNA for Ku70/80 and DNA-PKcs to bind, so DNA ends are peeled off from the histones. In the case of the 147 bp nucleosome, only one end is peeled off, resulting in partial binding by Ku70/80.

In addition, the authors analyzed potential effects of the histone variant H2AX, which replaces H2A on damaged DNA, on Ku70/80 binding, as well as the influence of DNA-PKcs and ATP on the combined binding of Ku70/80 and DNA-PKcs to these nucleosomes. These findings raise more questions than answers (see comments below). Due to the lack of structural knowledge regarding DSB repair in the context of nucleosomes, the results presented in this manuscript are novel and warrant consideration for publication.

Again, we thank the reviewer for noting that the results presented in this manuscript are novel and warrant consideration for publication.

However, several issues related to DNA substrate design and data interpretation should be addressed first. A scientist should recognize the limitations of each experiment and interpret the results based on experimental data and established scientific knowledge.

1. The three nucleosome DNAs used in this study as substrates for DSB repair are unrealistic. First, in a cellular context, two DNA breaks surrounding a nucleosome would likely lead to the loss of the nucleosome rather than its repair. Second, it is highly improbable that a DSB can occur so close to or even within a nucleosome, such as the 147 bp one, because a histone octamer would either slide off or move inward from the broken end, even if the DSB occurs immediately adjacent to a nucleosome, regardless of the presence of nucleosome remodelers. Third, the 601 Widom sequence binds tightly to a histone octamer and is not prone to sliding in either direction. This is evident from the structures of the 153 bp nucleosome-Ku70/80 complex, where only a limited number of DNA base pairs were “peeled” off at each end, and neither end results in optimal binding by Ku70/80.

We agree with the reviewer that the mono-nucleosomes used in this study do not represent the situation in the cell. Rather, those structures represent our progress in moving the field towards this eventual goal. To date, most of the structural work carried out on proteins binding to nucleosomes has been in the context of mono-nucleosomes^{3,4}. This is due to the high suitability of mono-nucleosomes for structural biology approaches compared to large nucleosome arrays and the significant advance in understanding of the molecular mechanisms that are afforded by these structures. We believe using a nucleosome array will only result in visualising Ku/DNA-PKcs bound to the DNA end as we show.

Secondly, the 147 bp construct, we agree is close/within a nucleosome, however this was used initially based on the publication⁵. This is why we increased the DNA length to 153 bp and eventually 197bp. The problem we faced when carrying out structural work, is that the longer the DNA length the greater the flexibility due to the distance of Ku70/80 and DNA-PKcs from the NCP as shown in the 197 bp structure, which is significantly lower resolution. We also believe that using a 147 bp nucleosome is not problematic and is a possible scenario. Particularly since it has been well documented that the histone octamer slides on the DNA and prefers to sit flush with the DNA ends⁶.

2. The authors observed that one end of the DNA in the 147-bp nucleosome is associated with Ku70/80 when a 4-fold excess of Ku70/80 is mixed with the nucleosome. Is the DNA end bound by Ku70/80 unique, or is it a mixture of both DNA ends? Did the histone core translocate along the DNA?

As shown in Supplementary Fig. 6a, and copied below, within the 2D classes we can see density corresponding to Ku70/80 (shown with red circles) on either side of the nucleosome. Therefore Ku70/80 is able to bind to both ends of the 147 bp nucleosome, and perhaps we are averaging a mixture of it binding to both ends when creating the ab initio/3D models. This is difficult to separate/distinguish in cryo-EM, but we have now added additional text to recognise this. We do not observe any translocation of the histone core along the DNA, as mentioned above the 601 Widom sequence binds very tightly.

3. How many base pairs are associated with Ku70/80 in the 147-bp nucleosome? Typically, a minimum of 14 bp are covered by one Ku70/80 molecule (PMID: 11493912). In the case of the 147 bp Widom nucleosome, approximately 140 bps are involved in histone binding. Peeling off any stretch of DNA to allow Ku70/80 to bind would be energetically costly. Ku70/80 binds free DNA ends with a dissociation constant of ~ 1 nM; however, the K_d of Ku70/80 when associated with nucleosomes appears to range from 50 to 100 nM, indicating it is 50-100 times weaker. Could such weak binding occur in vivo?

We thank the reviewer and agree a more detailed analysis of how Ku70/80 binds to DNA would be useful. Analysis of the DNA bases covered by Ku70/80 in the Ku70/80 + nucleosome structures have now been shown in **Supplementary Fig. 8**

The EMSA data we show is not enough to allow conclusion about the affinity of Ku70/80 in our system, however other techniques such as ITC, FIDA, BLI have not been possible due to various reasons as mentioned above and shown in **Figure 1- above**. Ku70/80 is a very abundant protein in the nucleus therefore if there are few other free DNA ends in the cell then a nucleosomal end can be expected to be bound very well.

4. The authors asserted multiple times that Ku70/80 is the first to arrive at broken DNA ends. What evidence do they have to support this statement? Recent measurements in living cells indicate that Ku70/80 and DNA-PKcs arrive at broken DNA ends simultaneously (PMID: 39578493). Furthermore, as quoted by the authors in reference #23, DNA-PKcs limits excessive loading of Ku70/80. If Ku70/80 arrives before DNA-PKcs, how does DNA-PKcs prevent the loading of multiple Ku70/80 onto a DNA end?

We thank the reviewer for this comment, the traditional understanding of the NHEJ mechanism was that Ku70/80 would arrive first. However, we agree that recent measurements indicate the simultaneous binding with DNA-PKcs. We have now edited the

text, to remove the statements that Ku70/80 binds first and appreciate the DNA-PK formation must occur as the reviewer mentions and referenced the paper above.

5. Human Ku70/80 typically slides along free DNA and does not remain at the DNA end in the absence of DNA-PKcs. Therefore, the observation of Ku70/80 remaining at the DNA ends on the 197-bp nucleosome is unusual. Is there steric hindrance preventing Ku70/80 from sliding toward the nucleosome? For studying Ku70/80 and DNA-PKcs binding, an ideal nucleosome may consist of 175 bp, leaving one free DNA end of 30 bp.

We do not observe any significant sliding; however, the resolution of this cryo-EM map is low, which we associate with flexibility, however it may indeed be due to Ku70/80 movement. We have now added some text to discuss this. We do not envisage the benefit of using further nucleosome DNA substrates for this publication as we are focusing on the initial recognition and synapsis and not ligation. However, we will consider using different DNA substrates for the subsequent NHEJ mechanism steps.

6. The cell data reporting SAP binding to chromatin is promising. However, it does not differentiate whether SAP binds to the acidic patch of the histone or the DNA, nor does it specify whether it interacts with H2A or H2AX. As a result, the implications of SAP binding to the acidic patch in the 153-bp nucleosome containing H2AX remain uncertain. Given the spatial constraint, it is possible that SAP docking on the histone acidic patch, which is common to both H2A and H2AX, could be coincidental.

We thank the reviewer for this comment. We should make it clear that the linker of Ku70 is what interacts with the acidic patch and that the SAP domain binds to the DNA. We have now adjusted the text to make this clear. The cell data confirms that it is the SAP domain which is important for binding and not the linker. When looking back at previous cryo-EM data, we do see some additional density for the SAP domain in other cryo-EM maps, however we were never able to resolve this. Therefore, H2AX may or may not be important for stabilisation of Ku70 and this is now discussed further and more openly in the manuscript with an additional figure (**Supplementary Fig. 14**).

7. As the acidic patch exists in both H2A and H2AX, have the authors measured the binding affinity of the SAP domain of Ku70 with H2A/H2B compared to H2AX/H2B? Additionally, how does SAP binding to the acidic patch compare to that of known binder of H2A, such as Dot1L, SAGA DUB or COMPASS?

We thank the reviewer for this idea, although after careful consideration, we believe this experiment would not give us any additional data. This is because the SAP domain binds to the DNA, therefore without the DNA present we don't believe the C-terminus of Ku70 would be stabilised on only histones. This has been shown by our inclusion of data from EpiCypher (**Supplementary Fig. 10**). We have now made it clear in the text that the interaction between the linker and the histone octamer is not the important interaction site, which we also discuss in the previous comment. We thank the reviewer for suggesting we compare the C-terminus of Ku70 interacting with the acidic patch with known binders of H2A. We have now added an additional Figure (**Supplementary Fig. 13**) and text to discuss the similarities and differences.

8. In the presence of DNA-PKcs, how many base pairs of the 153 bp nucleosome are bound by DNA-PKcs? A superimposition of the structure of DNA-PK bound to free DNA with that of the 153-bp nucleosome would be helpful. Only when 15 bp are bound to DNA-PKcs is the

kinase activated. Have the authors measured the DNA-PKcs kinase activity in the presence of 147- or 153-nucleosome?

We have now added more comparison figures (**Supplementary Figs. 7 and 24**) and shown how many DNA bps are covered by Ku70/80 and DNA-PKcs (**Supplementary Figs. 8, 23 and 26**). We have carried out phosphorylation gels in the presence of 153 bp nucleosomes and can see that the kinase activity is not active. However, for the purpose of this study to look at synapsis we believe this is not important. For future studies looking at transitions from long-range to short-range complexes we will extend the DNA length and check the ability to carry out DNA-PKcs kinase activity.

9. Given that the DNA end is blunt and open, did the authors observe specific interactions between DNA-PKcs and the DNA end, such as the DEB (DNA end blocking) helix of DNA-PKcs between the DNA strands? DEB binding to the DNA end may slow down but not completely inhibit DNA-PKcs kinase activity.

This is an interesting point; however, we do not observe the DEB helix in our data. Generally, we only observe this when LX4 is engaged.

10. Given the extremely short DNA length available for DNA-PKcs binding and the lack of evidence supporting DNA-PKcs activation, the conclusions regarding the roles of ATP and AMP-PNP in promoting DNA-PK dimerization and the “activated state” of DNA-PKcs are speculative, if not entirely unfounded.

We completely agree with the reviewer and have now carefully formulated these conclusions in the manuscript.

Reviewer #3:

The authors present cryo-electron microscopy structures of the human Ku70/80 heterodimer and DNA-PK holoenzyme bound to nucleosomes with varying DNA lengths, to investigate how non-homologous end joining machinery recognizes DNA double-strand breaks in chromatin. They show that Ku70/80 binding can unwrap nucleosomal DNA and find that the Ku70 SAP domain binds nucleosomal DNA in the Ku complex. Further interactions are found at the histone acidic patch and involve the linker region between the Ku80 core domain and the SAP domain. Additionally, they reveal a conformational change in Ku80 upon DNA-PK binding and demonstrate that non-hydrolysable ATP promotes DNA end synapsis. Altogether, these findings advance our understanding of the binding of Ku to chromatinized DNA and provide structural insights into DSB repair by NHEJ. All in all, the paper is well written and the structural analysis generally solid. The paper covers structural biology up to some cell based assays. The area of research is of interest to both basic and biomedical sciences. In general, I believe the manuscript is of interest to the readers of Nature Communications. However, there are some parts where I believe the evidence needs to be solidified before I can recommend publication.

We thank the reviewer for these comments that the data advances our understanding of binding Ku to chromatin, and that the structural analysis is solid.

Major points:

1) Line 171/ Line 179-180: Is the appearance of the extra density and the proposed stabilisation of the flexible tail specific to H2AX? This is somewhat surprising since the acidic patch is also present in canonical histones. Are there any differences between H2AX and H2A at contact points between Ku and the nucleosome? Or can altered DNA unwrapping account for the observed difference, and if so, would that be specific to H2AX? To validate the differences in binding, perhaps you could perform a competition experiment to see whether H2AX nucleosomes are bound better than H2A nucleosomes?

We thank the reviewer for this comment, and as mentioned above upon further analysis of our data we do observe some SAP density in the H2A structures, showing that the SAP is still able to bind. We do observe improved density with H2AX, which may be due to either the quality of the data or stabilisation through the extended tail. We have now added further text and **Supplementary Fig. 14** to show this. In terms of biochemical analysis, as described above we have found this aspect of the research very challenging with the nucleosomes due to various reasons mentioned above and shown in **Figure 1- above**. We are also unable to carry out experiments of only H2A/H2B and the SAP domain because this requires DNA binding within the context of the nucleosome. We have however added an additional EMSA gel (**Supplementary Fig. 17**) showing how the SAP domain regulates Ku loading to DNA.

2) What is the origin of the specific model shown in Figure 2E? The authors do not really show detailed density for that part and even state that they could not confidently identify specific histones contacts solely from cryo-EM maps. How can they confidently model the linker? Does AlphaFold predict this interaction? I am not yet convinced on the basis of the provided data for the physiological relevance of the SAP and linker nucleosome binding. Since this is a main part and conceptual advance in this manuscript, I suggest that the authors better characterise this interaction in vitro, e.g. using a construct that contains only linker and SAP domains and demonstrate nucleosome and acidic patch binding and work out any differences between H2AX and H2A nucleosomes.

The model shown in Figure 2E was from docking the known SAP domain PDB into density for the SAP and tracing density corresponding to the linker from the SAP domain up to Ku70. We agree with the reviewer, although we could trace density for the linker we cannot be confident of specific residues in exact locations, due to limited resolution for amino acid side chains. We have now modified this Figure to now show density corresponding to this region and removed stick representations of any residues within the linker. We thank the reviewer for asking whether AlphaFold predicts this interaction as we had not tested this. We have now added an additional Figures (**Supplementary Figs. 15 and 16**) and text to show how AlphaFold struggles to predict the exact location of the linker, although a very rough region can be identified. We also show that if the full protein and nucleosome interaction are predicted through AlphaFold, however, again AlphaFold cannot confidently predict where the SAP and linker would bind, with perhaps one of the top 5 ranks being close to what we observe experimentally. As mentioned above in response to similar comments, upon looking back at previous data we also observe density for the SAP domain in the H2A constructs. We have now added an additional Figure (**Supplementary Fig. 14**) and text to explain this. We also explain more clearly in the manuscript that the functional data shows it is the SAP domain rather than the linker which is important for chromatin recruitment. The linker interaction would be dependent on where the double-strand break would occur and is therefore more likely to be unspecific binding. We have now added more text for this. With this in mind to study interactions between Ku and nucleosomes would require full length

protein and nucleosome constructs. As mentioned in detail above we have tried several biochemical experiments including ITC, FIDA and BLI and all have proved unsuccessful (**Figure 1 – above**). Finally, we have explored the interaction between the SAP domain and DNA further by expression and purification and EMSA gels and unfortunately the SAP domain in isolation does not bind to DNA and therefore is difficult to study (**Figure 2 – above**).

Minor and editorial points:

Line 53: Perhaps define DNA-PK as DNA-dependent protein kinase first before abbreviating it.

We thank. The reviewer and have now changed this.

Line 161: Please indicate at which position DNA-PKcs phosphorylates H2AX.

We have now added this as Serine 139.

Figure 2 A and C : These panels overlap with gel labels (“Nucleosomes”) covered by Figure 2C.

We thank the reviewer as we had not noticed this and have now adjusted the figure accordingly.

Figure 2 and general: Consider standardising the use of capital vs. lowercase letters.

We have now checked and adjusted.

Extended Figure 2 : The local refinement depictions appear somewhat blurry (also in Extended Figure 5). In general, the processing scheme images are not very sharp. Perhaps this is due to the downsampled review files, but if not, please consider improving their resolution and contrast.

This has now been checked and replaced.

References

1. Walker, J. R., Corpina, R. A. & Goldberg, J. Structure of the Ku heterodimer bound to DNA and its implications for double-strand break repair. *Nature* **412**, 607–614 (2001).
2. Britton, S., Coates, J. & Jackson, S. P. A new method for high-resolution imaging of Ku foci to decipher mechanisms of DNA double-strand break repair. *J Cell Biol* **202**, 579–595 (2013).
3. Shioi, T. *et al.* Cryo-EM structures of RAD51 assembled on nucleosomes containing a DSB site. *Nature* **628**, 212–220 (2024).
4. Boyer, J. A. *et al.* Structural basis of nucleosome-dependent cGAS inhibition. *Science* **370**, 450–454 (2020).

5. Gerodimos, C. A., Watanabe, G. & Lieber, M. R. Nonhomologous DNA end joining of nucleosomal substrates in a purified system. *DNA Repair (Amst)* **106**, 103193 (2021).
6. Flaus, A., Luger, K., Tan, S. & Richmond, T. J. Mapping nucleosome position at single base-pair resolution by using site-directed hydroxyl radicals. *Proc Natl Acad Sci U S A* **93**, 1370–1375 (1996).

Response to Reviewers 2

We thank the editor and reviewers for their thorough and thoughtful evaluation of our manuscript. We are encouraged by the overall positive feedback and pleased that the reviewers consider our work a significant contribution to the field and suitable for publication. Below, we provide a detailed point-by-point response to the remaining comments.

REVIEWER COMMENTS

Reviewer #1 (Remarks to the Author):

The authors have thoroughly addressed my initial comments/concerns, with one exception. I still believe that the cellular experiments carried out to define the functional importance of the SAP-linker in chromatin binding are underdeveloped and difficult to interpret, as was also noted by the other reviewers. Given the inability to characterize any of the SAP-linker mutants *in vitro*, the use of a truncated SAP-linker for the cellular experiments, and the absence of DSBs in these cellular experiments, it's unclear how the findings connect back to the role of the SAP-linker in Ku70/Ku80 DSB recognition in chromatin that the authors thoroughly describe in their structural analysis. I do appreciate the technical difficulties the authors noted for these experiments, and so I feel it best to default to the editor to decide if this was sufficiently addressed and/or whether these experiments in their current state add any value to the manuscript. I must stress that the rest of this manuscript represents a significant advance in the DNA repair field, and I do feel the manuscript is otherwise suitable for publication in Nature Communications. I applaud the authors for taking on this challenging project.

We thank the reviewer for acknowledging that the majority of the initial comments and concerns have been addressed. In the revised manuscript, we have now further characterized key SAP-linker mutants *in vitro* and incorporated these new findings. Using EMSA assays, we demonstrate that the Ku70 SAP domain binds directly to DNA. Notably, mutation of the conserved K596 residue results in a marked reduction in DNA-binding affinity, highlighting its critical role in mediating this interaction. These biochemical findings support our cellular data, which also identified K596 as essential for efficient Ku70/80 recruitment to chromatin.

These results are now presented in an updated Figure 2, along with corresponding additions to the main text and Methods section. Furthermore, we provide additional EMSA data in Supplementary Figure 17, demonstrating the importance of T577 in SAP domain–DNA binding.

Minor Comments

1. The gel in Supplementary Figure 17 appears to be mislabeled. I believe the right side of the gel should be +Ku70/80-ΔSAP.

This has now been changed, thank you for noticing.

Reviewer #2 (Remarks to the Author):

The revised manuscript includes changes in wording, but it lacks any new experiments necessary to assess the significance of the structural observations made with the Ku70/80 and nucleosome complexes. ED Fig. 17 does not present the binding affinities of full-length Ku70/80 and the Δ SAP variant to free DNA; instead, it demonstrates the loading of multiple Ku70/80 molecules onto a 153-bp oligo DNA. Additionally, there are no measurements of the binding affinity of Ku70/80 to nucleosomes (147 bp or 153 bp) as requested in the first round of review.

The following measurements are necessary for the reconsideration of the manuscript:

- (a) K_d of Ku70/80-FL and Ku70/80- Δ SAP to free DNA oligonucleotides, e.g., 20 bp.
- (b) K_d of Ku70/80-FL and Ku70/80- Δ SAP to 147-bp and 153-bp nucleosomes.

To avoid loading of multiple Ku70/80 molecules onto one DNA duplex, K_d can be measured with binding partners at a constant molar ratio of 1:1.

We thank the reviewer for raising the important point regarding the binding affinities of Ku70/80 to DNA, both in the presence and absence of the SAP domain. This topic has recently been addressed in the publication titled "*The KU70-SAP domain has an overlapping function with DNA-PKcs in limiting the lateral movement of KU along DNA*", in which one of our co-authors, Dr. Mauro Modesti, is also a contributor. After discussion within our team, we concluded that replicating the same experiments would not significantly enhance the value of our current study.

Nonetheless, to strengthen the manuscript, we have incorporated an additional experiment in collaboration with Dr. Modesti. This new experiment, now included in Figure 2 and Supplementary Figure 17, presents an EMSA analysis showing the binding of the wild-type Ku70 SAP domain to DNA. Importantly, we also demonstrate the impact of two specific mutations (K596A and T577E) in the SAP domain, which result in reduced DNA binding, thereby underscoring the functional relevance of these residues.

We believe this new data provides valuable mechanistic insight into the DNA-binding activity of the Ku70 SAP domain. As noted in our previous response, attempts to determine K_d values for binding to nucleosomes have proven technically challenging, and we were unfortunately unable to obtain reliable measurements.

Reviewer #3 (Remarks to the Author):

The authors satisfactorily addressed my points. I appreciate efforts to better

understand SAP domain nucleosome binding and the manuscript is revised appropriately. I am happy with the current version and can support publication.

Many thanks for your response.

Response to Reviewers 3

We thank the editor and reviewers for their continued thorough and thoughtful evaluation of our manuscript. We have added our comments below and now hope the manuscript is ready for publication.

Reviewer #1 (Remarks to the Author):

The authors have addressed my remaining concerns.
Many thanks.

Reviewer #2 (Remarks to the Author):

The EMSA data published or presented in this manuscript support the conclusion that the SAP domain binds to DNA; however, it does not provide evidence for binding to histones. Furthermore, the mutant analysis reinforces the importance of DNA binding by the SAP domain, but it does not address whether SAP binds to chromatin proteins, contrary to the implications made by the authors.

The manuscript lacks data to substantiate the claim that Ku70/80 binds to nucleosomes beyond its interaction with DNA.

The reviewer is correct; the SAP domain binds DNA not histones – we have added an additional panel to Fig 2F to show it can also bind nucleosome bound DNA and free DNA. We have also modified the text throughout to make this clear.

The revised manuscript includes changes in wording, but it lacks any new experiments necessary to assess the significance of the structural observations made with the Ku70/80 and nucleosome complexes. ED Fig. 17 does not present the binding affinities of full-length Ku70/80 and the Δ SAP variant to free DNA; instead, it demonstrates the loading of multiple Ku70/80 molecules onto a 153-bp oligo DNA. Additionally, there are no measurements of the binding affinity of Ku70/80 to nucleosomes (147 bp or 153 bp) as requested in the first round of review.

The following measurements are necessary for the reconsideration of the manuscript:

- (a) K_d of Ku70/80-FL and Ku70/80- Δ SAP to free DNA oligonucleotides, e.g., 20 bp.
- (b) K_d of Ku70/80-FL and Ku70/80- Δ SAP to 147-bp and 153-bp nucleosomes.

To avoid loading of multiple Ku70/80 molecules onto one DNA duplex, K_d can be measured with binding partners at a constant molar ratio of 1:1.

The EMSA data published or presented in this manuscript support the conclusion that the SAP domain binds to DNA; however, it does not provide evidence for binding to histones. Furthermore, the mutant analysis reinforces the importance of DNA binding by the SAP domain, but it does not address whether SAP binds to chromatin proteins, contrary to the implications made by the authors.

The manuscript lacks data to substantiate the claim that Ku70/80 binds to nucleosomes beyond its interaction with DNA.